# Genetic deficiency of NOD2 confers resistance to invasive aspergillosis

Mark S. Gresnigt [1,2,3], Cristina Cunha[4,5], Martin Jaeger[2], Samuel M. Gonçalves[4,5], R.K.Subbarao Malireddi [6], Anne Ammerdorffer [2], Rosalie Lubbers[2], Marije Oosting[2], Orhan Rasid [1], Grégory Jouvion[7], Catherine Fitting[1], Dirk J. de Jong[8], João F. Lacerda[9,10], António Campos Jr.[11], Willem J.G. Melchers [12], Katrien Lagrou [13,14], Johan Maertens[13,15], Thirumala-Devi Kanneganti[6], Agostinho Carvalho [4,5], Oumaima Ibrahim-Granet[1] & Frank L. van de Veerdonk[2]

Invasive aspergillosis (IA) is a severe infection that can occur in severely immunocompromised patients. Efficient immune recognition of *Aspergillus* is crucial to protect against infection, and previous studies suggested a role for NOD2 in this process. However, thorough investigation of the impact of NOD2 on susceptibility to aspergillosis is lacking. Common genetic variations in *NOD2* has been associated with Crohn's disease and here we investigated the influence of these genetic variations on the anti-*Aspergillus* host response. A *NOD2* polymorphism reduced the risk of IA after hematopoietic stem-cell transplantation. Mechanistically, absence of NOD2 in monocytes and macrophages increases phagocytosis leading to enhanced fungal killing, conversely, NOD2 activation reduces the antifungal potential of these cells. Crucially, *Nod2* deficiency results in resistance to *Aspergillus* infection in an *in vivo* model of pulmonary aspergillosis. Collectively, our data demonstrate that genetic deficiency of *NOD2* plays a protective role during *Aspergillus* infection.

[1] Cytokines & Inflammation, Institut Pasteur, 28 rue du Dr Roux, 75724 Paris, France. [2] Department of Experimental Internal Medicine and Radboud Center for Infectious Diseases (RCI), Radboud University Medical Center, Geert Grooteplein zuid 8, 6525GA Nijmegen, The Netherlands. [3] Department of Microbial Pathogenicity Mechanisms, Leibniz Institute for Natural Product Research and Infection Biology, Hans Knöll Institute, Beutenbergstraße 11a, 07745 Jena, Germany. [4] Life and Health Sciences Research Institute (ICVS), School of Medicine, University of Minho, Campus de Gualtar, 4710-057 Braga, Portugal. [5] ICVS/3B's - PT Government Associate Laboratory, Braga/Guimarães, Campus de Gualtar, 4710-057 Braga, Portugal. [6] Department of Immunology, St. Jude Children's Research Hospital, 262 Danny Thomas Place, Memphis, TN 38105, USA. [7] Département Infection et Epidémiologie, Unité Histopathologie Humaine et Modèles Animaux, Institut Pasteur, 28 rue du Dr Roux, 75724 Paris, France. [8] Department of Gastroenterology and Hepatology, Radboud University Medical Center, Geert Grooteplein zuid 8, 6525GA Nijmegen, The Netherlands. [9] Instituto de Medicina Molecular, Faculdade de Medicina de Lisboa, Universidade de Lisboa, Av. Professor Egas Moniz, 1649-028 Lisboa, Portugal. [10] Serviço de Hematologia e Transplantação de Medula, Hospital de Santa Maria, 1649-035 Lisboa, Portugal. [11] Serviço de Transplantação de Medula Óssea (STMO), Instituto Português de Oncologia do Porto, Rua Dr. António Bernardino de Almeida, 4200-072 Porto, Portugal. [12] Department of Medical Microbiology, Radboud University Medical Centre, Nijmegen, the Netherlands, Geert Grooteplein zuid 8, 6525GA Nijmegen, The Netherlands. [13] Department of Microbiology and Immunology, KU Leuven, Herestraat 49 Box 1030, 3000 Leuven, Belgium. [14] Department of Laboratory Medicine and National Reference Center for Medical Mycology, University Hospitals Leuven, Herestraat 49 Box 1030, 3000 Leuven, Belgium. [15] Department of Hematology, University Hospitals Leuven, Herestraat 49 Box 1030, 3000 Leuven, Belgium. These authors jointly supervised this work: Oumaima Ibrahim-Granet, Frank L. van de Veerdonk. Correspondence and requests for materials should be addressed to Frank L. van de Veerdonk. (email: Frank.vandeveerdonk@Radboudmc.nl)

Humans are ubiquitously exposed to airborne spores of *Aspergillus*, but only severe immunocompromised patients are at risk of developing pulmonary invasive aspergillosis (IA)[1]. Patients undergoing hematopoietic stem cell transplantation (HSCT) have a distinctive elevated susceptibility to aspergillosis and infections in these patients are associated with a high mortality[2]. With increasing knowledge of the antifungal host response, it has become evident that not only the immunocompromised status of patients plays a role in susceptibility to infection, but also the genetic background of both the engrafted bone marrow and the recipient[3]. To provide a good risk stratification for the development of IA following HSCT, genetic susceptibility needs to be taken into account. Common polymorphisms in various pattern recognition receptor (PRR) pathways are known to be associated with an increased risk for aspergillosis, which includes dectin-1[3,4], pentraxin-3[5] as well as many other receptors[6]. These findings help to predict susceptibility, yet also provide insight into the importance of these pathways in antifungal host defence. These studies, therefore, may also aid in the development of novel immune targeted treatment strategies. Nevertheless, several immune pathways remain unexplored for susceptibility to aspergillosis.

One of these relatively unexplored receptor families are the NACHT-LRR receptors (NLRs), to which the intracellular nucleotide-binding oligomerization domain (NOD) receptors belong. The NLR receptor Nlrp3 plays a role in IA via regulation of inflammasome activation and subsequently protective IL-1-mediated cytokine responses[7]. Two other NLR receptors NOD1 and NOD2 are primarily involved in the recognition of peptidoglycan-derived moieties from bacteria and in the induction of proinflammatory host responses[8–12]. Although *Aspergillus* does not contain peptidoglycan, some evidence suggests that these NOD receptors might play a role in host defence against aspergillosis[13–16]. In contrast to a previous study that demonstrated a crucial role for NOD1 in activation of corneal epithelial cells by *Aspergillus*[16], we recently reported that NOD1 negatively modulates host defence by reducing cytokine responses and oxidative burst[17]. NOD2, another member of the NLR family is highly expressed in lungs of mice infected with *Aspergillus*, and in THP1 cells, RAW macrophages and A549 cells stimulated with *Aspergillus*[13,14]. The NOD2 agonist Muramyl-dipeptide (MDP) can synergistically increase *Aspergillus*-induced cytokine levels[13,14]. NOD2 also may play a role in host defence against fungal keratitis[15,16]. Recently, NOD2 was also suggested to play a role in the recognition of chitin[18], a polysaccharide that is present in the cell wall of all fungi. Polymorphisms in *NOD2* have been associated with host defence against infectious diseases. In particular, strong associations between *NOD2* polymorphisms and susceptibility to tuberculosis have been identified[19–22]. It should, however, be noted that the strongest genetic association with *NOD2* is with Crohn's disease[23]. Polymorphisms in *NOD2* impact autophagy and antigen presentation in host defence against bacteria[24]. Considering recent evidence suggesting a crucial role

for the autophagy machinery in host defence against *Aspergillus*[25], *NOD2* is a candidate susceptibility gene for aspergillosis.

Although previous studies have revealed enhanced expression of *NOD2* during aspergillosis and synergism with *Aspergillus*-induced cytokine responses, these studies did not thoroughly investigate whether NOD2 modulates anti-*Aspergillus* host defence. Additionally, it is unknown whether defective NOD2 signalling influences susceptibility to aspergillosis. Therefore, the present study investigates whether common polymorphisms in *NOD2* or its complete deficiency influences susceptibility to IA, and whether it affects the immune response to *Aspergillus*. We demonstrate that genetic variation in *NOD2* in humans and complete *Nod2* deficiency in mice protects against IA. In line with this, NOD2 deficiency or its neutralization associates with increased antifungal activity of macrophages and monocytes, conversely NOD2 activation neutralizes fungal killing capacity of phagocytes. Our data collectively highlight a detrimental role for NOD2 receptor in anti-*Aspergillus* host defence.

## Results

### *NOD2* genetic variation decreases the risk of IA after HSCT.

To investigate the relationship between genetic variability in *NOD2* and susceptibility to IA, four nonsynonymous SNPs in the *NOD2* coding sequence were analysed (Table 1). The probability of IA was assessed according to recipient or donor genotypes by estimating the cumulative incidence of infection among transplant recipients at 24 months after HSCT. Among the SNPs tested, the donor, but not recipient, P268S (rs2066842) SNP in *NOD2* was associated with an increased risk of IA (Fig. 1a). Other polymorphisms did not allow accurate risk estimations due to low (<0.05) allele frequencies in our study population. The cumulative incidence of IA for donor P268S was 32.7% for the CC genotype, 21.6% for CT and 20.0% for TT genotypes, respectively (Fig. 1a). The key contribution of the CC genotype to the risk of infection was further illustrated upon modelling a dominant mode of inheritance (cumulative incidence of IA, 32.7% for CC vs. 21.3% for CT and TT genotypes combined) (Fig. 1b). In a multivariate model accounting for age, gender, post-transplant neutropenia and acute graft-versus-host disease (GVHD)[26], the donor CC genotype at P268S conferred a 2.1-fold increased risk of developing IA after transplantation (95% CI, 1.15–4.47; $p = 0.021$; $p$-values calculated using Gray's test). Collectively, these results highlight genetic variation at the *NOD2* locus as a critical risk factor regulating susceptibility to IA after HSCT.

### The *NOD2* P268S SNP alters pulmonary cytokine levels in IA.

To assess whether the genotypes at P268S in *NOD2* differentially regulate pulmonary inflammation in aspergillosis, cytokine levels in bronchoalveolar lavage (BAL) samples from patients with IA were assessed. Genotype-specific differences were observed, with patients transplanted with bonemarrow carrying the TT genotype displaying lower median concentrations of IL-10 and IL-8 than

**Table 1 Description of the SNPs in the *NOD2* gene evaluated in our study**

| RefSNP | Genome coordinates | aa change | Alleles | CEU MAF | MAF in our study | HWE |
|--------|---------------------|-----------|---------|---------|------------------|-----|
| rs2066842 | chr16:50710713 | P268S | C > T | 0.102 | 0.278 | 0.72 |
| rs2066844 | chr16:50712015 | R702W | C > T | 0.014 | 0.027 | 0.77 |
| rs2066845 | chr16:50722629 | G908R | G > C | 0.005 | 0.002 | 1.00 |
| rs2066847 | chr16:50729867 | 1007fs | — > C | 0.006 | 0.022 | 0.98 |

Publically available sequencing data from Pilot 1 of the 1000 Genomes Project (www.1000genomes.org) was used to determine MAF. Genome coordinates were extracted from the hg18 build
*SNP* single-nucleotide polymorphism, *aa* amino acid, *P* prolins, *S* serine, *R* arginine, *W* tryptophan, *G* Glycine, *fs* frameshift, *CEU* Utah Residents (CEPH) with Northern and Western Ancestry, *MAF* minor allele frequency, *HWE* Hardy Weinberg Equilibrium

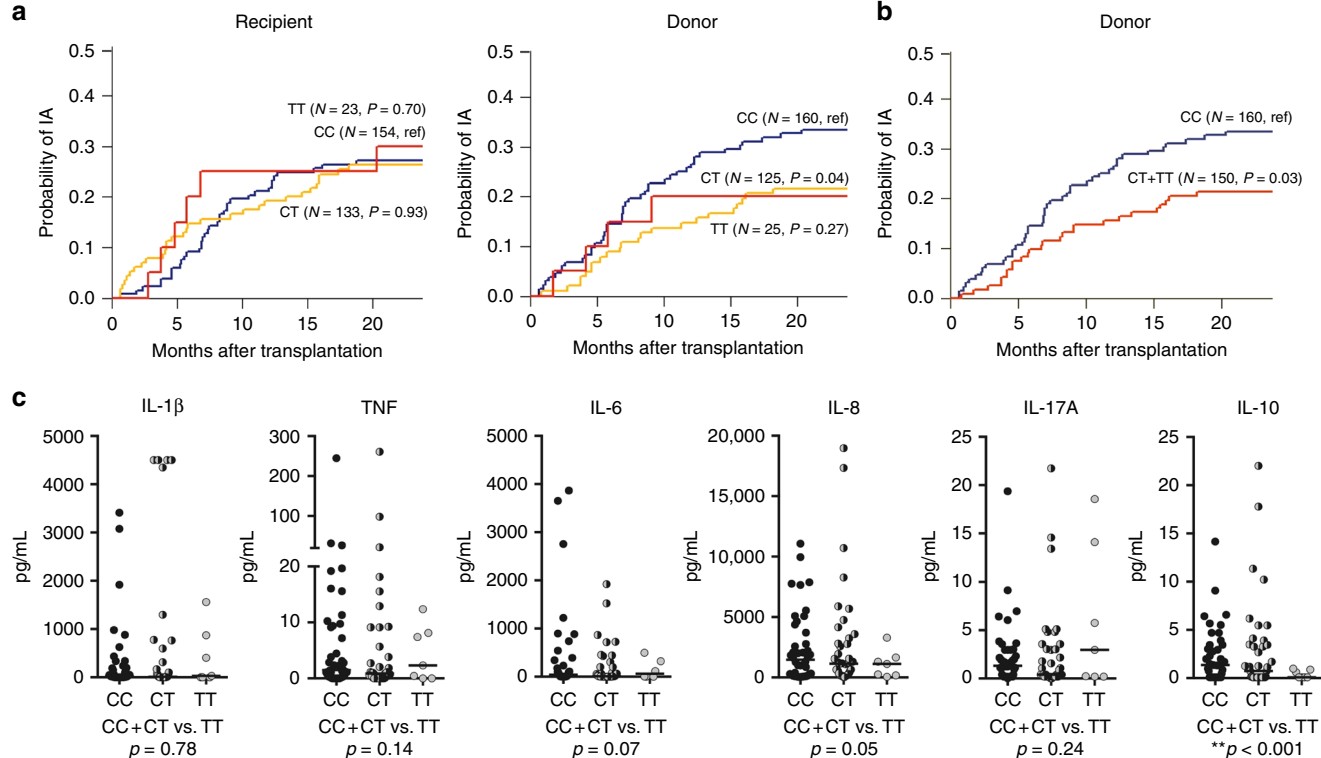

**Fig. 1** Genetic variation in donor P268S (rs2066842) confers resistance to IA after HSCT. Shown are the results obtained in a cohort comprising 310 eligible patients and respective donors. Cumulative incidence of IA according to (**a**) donor or recipient genotypes at rs2066842 or **b** a dominant genetic model of donor genotypes at rs2066842. In **a**, the blue line represents the carriers of the reference (ref) CC genotype, the red line carriers of the heterozygous CT genotype, and the yellow line represents carriers of the homozygous TT genotype. In the dominant model (**b**) red line represents the carriers of both the CT and TT genotypes. Data were censored at 24 months, and relapse and death were considered competing events. *p*-values were calculated using Gray's test. **c** IL-1β, TNF, IL-6, IL-8, IL-17A, and IL-10 levels measured in the BAL of patients with aspergillosis and stratified based on the *NOD2* rs2066842 donor genotypes. Each dot represent an individual patient, with black filled dots representing HSCT donor carriers of the reference CC genotype, half filled black/gray dots representing carriers of the heterozygous CT genotype, and gray dots representing carriers of the homozygous TT genotype. Data are represented scatter dot plot with the median; with *p*-values were calculated using the Mann–Whitney *U* test, *p*-values of statistical tests are shown within the graphs

CC + CT carriers (0.41 vs. 2.6 pg/mL; and 1125 vs. 2560 pg/mL). In addition, a trend toward decreased IL-6 and TNF levels was also observed among patients transplanted with bonemarrow from TT carriers (Fig. 1c). Collectively, these findings point to a *NOD2* genotype-determined alteration in cytokine production in response to *Aspergillus* infection.

**NOD2 variants alter A. fumigatus-induced cytokine responses.**
To examine the impact of *NOD2* variation on host defence against *Aspergillus*, the impact of the *NOD2* genetic variants on *Aspergillus* immune recognition and cytokine production was investigated. In vitro *Aspergillus*-induced cytokine responses of primary human PBMCs were stratified based on P268S (rs2066842), G908R (rs2066845), and R702W (rs2066844) genotypes, to investigate their influence on cytokines responses to *A. fumigatus*.

Individuals carrying the T-allele at P268S, which was associated with a reduced susceptibility to aspergillosis in patients (Fig. 1), induced significantly lower IL-1β and demonstrated a trend toward lower TNF production in response to *Aspergillus* stimulation (Fig. 2a). Additionally, the TT-genotype was associated with significantly lower IL-17A responses compared with individuals carrying the CC or CT genotypes (Fig. 2b). Heterozygous carriers of the G908R and R702W polymorphisms did not show significantly altered cytokine responses to

*Aspergillus* (Supplementary Fig. 1), and homozygous carriers of these polymorphisms were not represented within our cohort.

Insertion of a cysteine at position 1007 (1007finsC) (rs2066847) induces a frameshift, which results in a defective NOD2 receptor, and homozygous carriage of this mutation results in complete NOD2 deficiency and is highly associated with Crohn's disease[27]. Healthy individuals heterozygous for this mutation demonstrated significantly lower IL-1β and a trend toward lower TNF, but not IL-6 responses to *Aspergillus* (Fig. 2c). The decreased IL-1β correlated with significantly lower IL-17A responses in individuals carrying the Cysteine-insertion on one allele (Fig. 2d). Interestingly, *Aspergillus*-induced IFNγ or IL-22 production was not affected by this genotype (Fig. 2d).

**The 1007insC frameshift mutation enhances fungal killing.** For the P268S and 1007insC polymorphisms the impact on fungal killing capacity was evaluated. Although, the P268S polymorphism did not significantly impact *Aspergillus* killing (Fig. 2e), PBMCs of healthy individuals carrying the Cysteine-insertion (rs2066847) on one allele had a significantly increased capacity to neutralize *Aspergillus*-conidia (Fig. 2f). Production of reactive oxygen species (ROS) is highly important for host defence against aspergillosis, especially when considering that CGD patients that cannot produce ROS are highly susceptible to aspergillosis[28]. However, oxidative burst in response to *Aspergillus* was not influenced by the different *NOD2* genotypes (Fig. 2g, h).

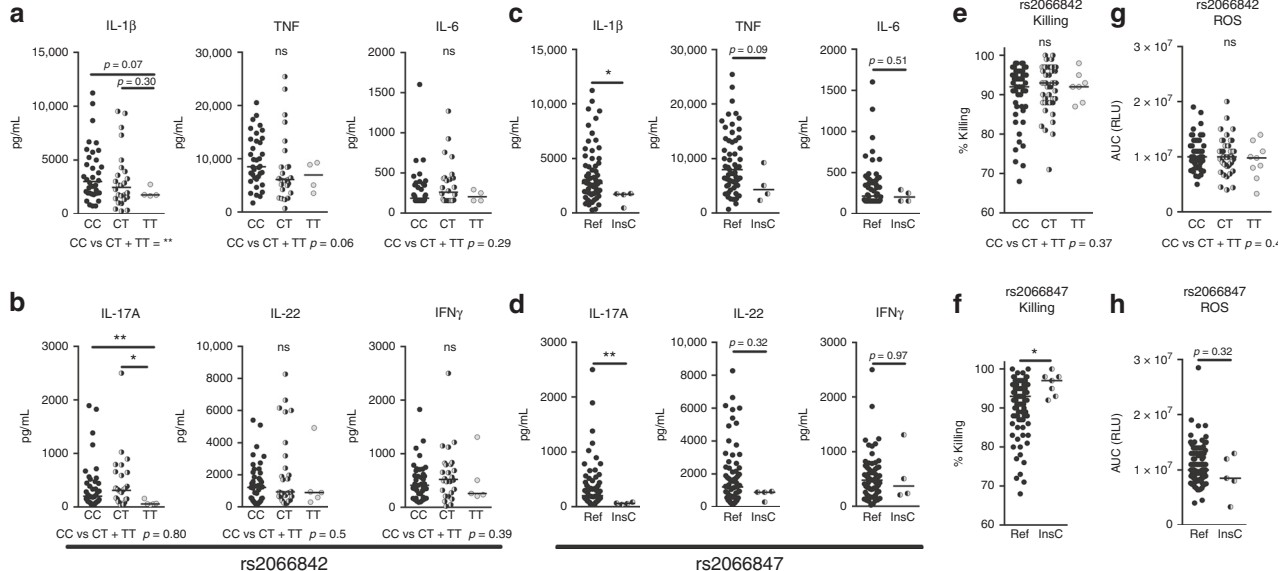

**Fig. 2** Human *NOD2* polymorphisms influence *Aspergillus*-induced cytokine responses and fungal killing. **a–d** IL-1β, TNF, IL-6, IL-17A, IL-22, and IFNγ levels measured in culture supernatants of PBMCs stimulated with (**a, c**) live *Aspergillus* conidia for 24 h or (**b, d**) heat-inactivated (HI) *Aspergillus* conidia for 7 days. The PBMCs of individuals with various genotypes of the *NOD2* gene were compared. These genotypes included (**a, b**) the P268S mutation (rs2066842; reference: CC $n = 36$, heterozygous: CT $n = 28$ and homozygous: TT $n = 4$) and (**c, d**) the 1007finsC mutation (rs2066847; reference $n = 62$ and heterozygous: insC $n = 4$). **e, f** Fungal killing capacity of human PBMCs assessed as CFU remaining of *A. fumigatus* ($2 \times 10^6$) following exposure for 24 h to ($5 \times 10^5$) PBMCs results are stratified based on the **e** P268S (rs2066842; ref: CC $n = 49$, heterozygous: $n = 45$ and homozygous: TT $n = 7$) and (**f**) 1007finsC (rs2066847; ref $n = 98$ insC $n = 7$) genotypes. **g, h** Area under the curve (AUC) of relative light units (RLU) induced by luminol oxidation by reactive oxygen species (ROS) released by PBMCs, results are stratified based on the (**g**) P268S (rs2066842; reference: CC $n = 47$, heterozygous: $n = 50$ and homozygous: TT $n = 9$) and (**h**) 1007finsC (rs2066847; ref $n = 112$ insC $n = 5$) genotypes. Data are represented scatter dot plot with the median. Each dot represent an individual patient, with (**a, b, e, g**) black filled dots representing carriers of the ancestral (reference) CC genotype, half-filled black/gray dots representing carriers of the heterozygous CT genotype, and gray dots representing carriers of the homozygous TT genotype, and (**c, d, f, h**) black filled dots representing carriers of the reference (ref) genotype without insertion and half-filled black/gray dots representing carriers of one Cysteine insertion (insC). The means were compared using the Mann–Whitney *U* test, *p*-values of statistical tests are shown within the graphs

**Human *NOD2* deficiency reduces *Aspergillus*-induced cytokines.** To further investigate the importance of NOD2 in *Aspergillus*-induced cytokine response, we analysed responses of primary human PBMCs within a background of complete NOD2 deficiency. PBMCs from patients with Crohn's disease, homozygous for the 1007finsC polymorphism and thus deficient for the NOD2 receptor, were stimulated with *A. fumigatus*. NOD2-deficient PBMCs demonstrated significantly lower IL-1β and TNF responses compared to controls (Fig. 3a). NOD2 deficient PBMCs also showed a significant reduction in production of the T-helper cytokines IL-22 and Interferon(IFN)γ and a trend toward decreased IL-17A induced by conidia (Fig. 3b). These reduced cytokine responses correlated with a reduced capacity to expand populations of IL-17A⁺, IL-22⁺, and IFNγ⁺ CD4 T-cells (Fig. 3c). Similar to individuals with heterozygous 1007insC mutations, the homozygous individuals demonstrated a trend toward improved fungal killing (Fig. 3d). However, no change in the capacity to induce ROS by zymosan or *Aspergillus* was observed (Fig. 3e).

***Nod2*⁻/⁻ mice are less susceptible to IA.** Since *NOD2* genetic variation was associated with a reduced risk of IA, the impact of full *Nod2* deficiency on susceptibility to aspergillosis was validated in an experimental model of IA. Wild-type (WT) C57BL/6 and *Nod2*-deficient (*Nod2*⁻/⁻) C57BL/6 mice were immunosuppressed using cyclophosphamide and subsequently subjected to lethal *Aspergillus* infection[29]. *Nod2*⁻/⁻ mice demonstrated a significantly improved 14-day survival, compared to WT mice (Fig. 4a). During infection, WT mice decline in bodyweight and seven out of eleven mice did not survive the infection whereas

eight out of nine *Nod2*⁻/⁻ mice survived the infection, despite having similar weight loss as WT mice during the first 3 days of infection (Fig. 4b). Although *Nod2*⁻/⁻ mice demonstrated severe symptoms such as hunching, head tilting, and circling, symptoms that have been described in the in vivo aspergillosis mouse model[30], they survived the infection, in contrast to WT mice. Bioluminescence imaging revealed that *Nod2*⁻/⁻ mice rapidly cleared the luciferase-expressing *Aspergillus*, whereas WT mice developed a fungal infection as indicated by a significantly higher luminescence signal on day 3 post infection (pi) (Fig. 4c, d). After day 3 pi the luminescence could not be reliably compared between groups due to mice dropping out of the experiment (Supplementary Fig. 2), and severe hypoxia in critically ill mice that influences bioluminescence readout. For assessment of histopathological damage in the lungs, inflammation and fungal burden, mice were sacrificed on day 3 pi. The decreased bioluminescence signal in the lungs of *Nod2*⁻/⁻ mice correlated with the fact that almost no *Aspergillus* DNA could be detected in the lung homogenates of *Nod2*⁻/⁻ mice (3 out of 8 mice were PCR positive with low values). However, in the lung homogenates of WT mice, *Aspergillus* PCR was positive for 5 out of 8 mice (Fig. 4e).

***Nod2*⁻/⁻ mice show reduced pulmonary histopathological damage.** Using histopathological analysis differences inflammatory and pathological damage to the lungs and sinuses of WT and *Nod2*⁻/⁻ mice were assessed. Within the lungs, WT mice displayed multifocal large areas of ischaemic necrosis (Fig. 5a I, circles and arrowheads), with fibrinous thrombi and destruction of blood vessels (Fig. 5a II, arrow). In contrast, *Nod2*⁻/⁻ mice

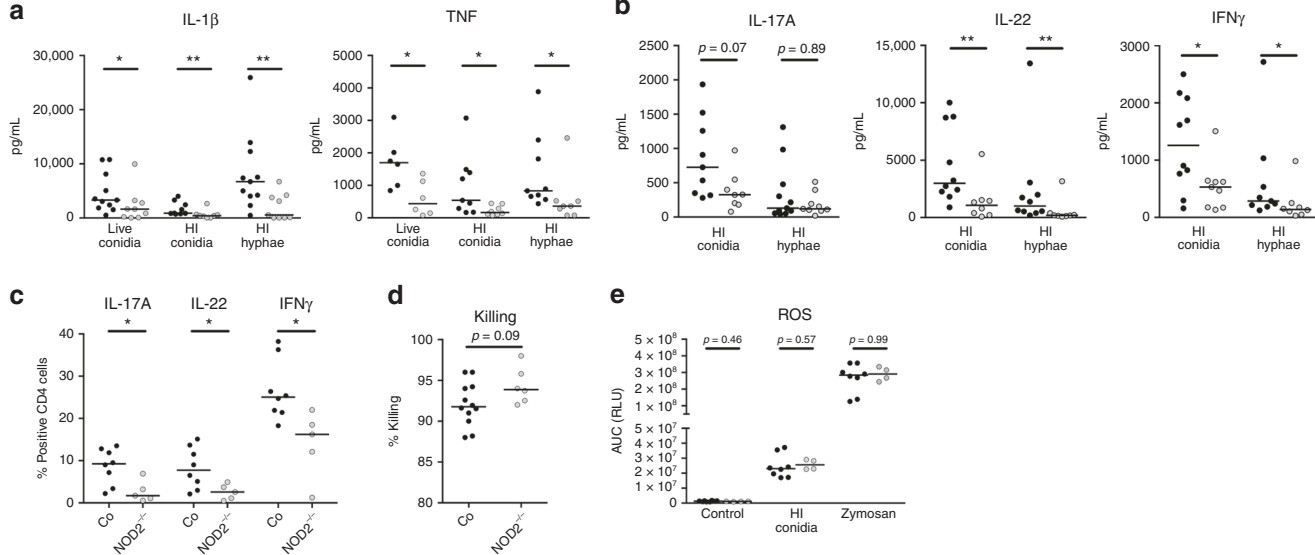

**Fig. 3** NOD2-deficient individuals have impaired cytokine responses in response to Aspergillus. **a** IL-1β and TNF levels in culture supernatants of PBMCs from healthy controls (black dots) and NOD2-deficient patients (gray dots) that were stimulated for 24 h with A. fumigatus live conidia, HI conidia or HI hyphae ($n = 11$ controls and $n = 9$ patients for IL-1β and $n = 9$ controls and $n = 8$ patients for TNF). **b** IL-17A, IL-22, and IFNγ levels after 7 days of stimulation with HI conidia or HI hyphae (IL-17A $n = 7$ controls and $n = 6$ patients; IL-22 and IFNγ $n = 10$ controls, $n = 8$ patients). **c** IL-17A+, IL-22+, and IFNγ+ CD4 T-cell populations ($n = 8$ controls and $n = 5$ patients) after 7 days stimulation with HI conidia shown as the percentage of total CD4 cells. **d** The fungal killing capacity of PBMCs ($5 \times 10^5$) from healthy controls and NOD2-deficient patients ($n = 12$ controls, $n = 6$ patients) counted as CFU remaining after 24-h stimulation with live A. fumigatus ($2 \times 10^6$). **e** The area under the curve of the reactive oxygen species release of PBMCs from healthy controls and NOD2-deficient patients (controls $n = 8$, patients $n = 4$) in response to live Aspergillus conidia ($1 \times 10^7$/mL) and zymosan (150 μg/mL) measured by luminescence signal from luminol conversion over 1 h. Data are represented as scatter dot plot with median and means were compared using the Mann–Whitney U test

rarely displayed inflammatory infiltrates and when present small (Fig. 5a I, circle and arrowhead; II, arrow). Severe fungal invasion was observed in WT mice (Fig. 5a III) with a high density of hyphae that invaded blood vessels (Fig. 5a III arrow). In contrast, fungi were rarely observed in lungs of $Nod2^{-/-}$ mice, only conidia (Fig. 5a III arrow), without invasion of the parenchyma or blood vessels. Macrophages were observed in the lesions, either randomly distributed (Fig. 5a IV, wild-type mice) or gathered in the small infiltrates (Fig. 5a IV, $Nod2^{-/-}$ mice). Although the immune suppression drastically decreased the number of F4/80+ cells no differences were observed between the groups (Fig. 5a IV).

Using morphometric analysis, the average number of lesions per section and the affected area was quantified. WT mice had a trend towards a higher average number of lesions per section (Fig. 5b). Moreover, the affected area of the lesions was significantly larger in WT mice (Fig. 5b).

Additional histology slides confirmed our morphometric analysis, as WT mice displayed marked lung lesions characterized by large foci of ischaemic necrosis (Fig. 5c I: left of the black line) with destruction of the bronchi/bronchiole epithelium (Fig. 5c I, black arrowheads), fungal invasion of lung parenchyma (Fig. 5c II top row), destruction of alveoli (Fig. 5c III top row) and invasion of blood vessels (Fig. 5c IV top row). Similar lesions were observed in other WT mice (Fig. 5c second row) with invasion of blood vessels (Fig. 5c III second row) and hyphae crossing the bronchiole epithelium lining (Fig. 5c IV second row, black arrowhead). In contrast, $Nod2^{-/-}$ mice displayed no or minimal lesions (Fig. 5c third and fourth row). At a low magnification, no lesions could be observed (Fig. 5c I, II fourth row), whereas at a high magnification), few hyphae could be detected in the alveoli/alveolar walls (Fig. 5c III, IV fourth row, black arrowheads).

WT mice displayed invasive sinusitis, whereas nasal sinus lesions were absent in $Nod2^{-/-}$ mice (Fig. 6). Arrowheads indicate destruction of nasal mucosa (Fig. 6a, enlarged in Fig. 6b), and invasion of fungi (Fig. 6c). Nasal sinuses of $Nod2^{-/-}$ mice did not demonstrate any signs of destruction (Fig. 6d, e) or presence of fungi (Fig. 6f).

**NOD2 augments Aspergillus-induced cytokine responses.** Since NOD2 genetic variation and its complete deficiency correlated with a decreased cytokine release, the capacity of NOD2 signalling to boost Aspergillus-induced cytokine responses was investigated. Co-stimulation of NOD2 by MDP augmented Aspergillus-induced IL-1β and TNF responses (Fig. 7a). This could, however, not be achieved in cells of Crohn's disease patients carrying the 1007finsC mutation (Fig. 7b). Similarly, cytokine responses to A. fumigatus by cells of $Nod2^{-/-}$ mice were investigated. Although BMDMs of $Nod2^{-/-}$ mice did not demonstrate altered IL-6, KC and TNF responses (Fig. 7c), splenocytes of $Nod2^{-/-}$ mice showed a reduced capacity to mount IL-6, KC, and TNF responses (Fig. 7d)

**NOD2 inhibits phagocytosis and killing of A. fumigatus.** The reduced susceptibility of $Nod2^{-/-}$ mice and patients with NOD2 genetic variants may be explained by enhanced killing capacity of myeloid cells due to their NOD2 deficiency, as monocytes from NOD2-deficient individuals demonstrated a trend toward Aspergillus killing (Fig. 3d). BMDMs from WT and $Nod2^{-/-}$ mice were compared for their fungal killing capacity, and $Nod2^{-/-}$ BMDMs proved to be more efficient at eradicating live Aspergillus conidia (Fig. 8a). Subsequently, NOD2 gene expression was silenced in human monocyte-derived macrophages (MDMs) to validate that the absence of NOD2 also positively influences fungal killing in human cells. Treatment of MDMs with NOD2 targeting siRNA augmented fungal killing capacity (Fig. 8b). Since

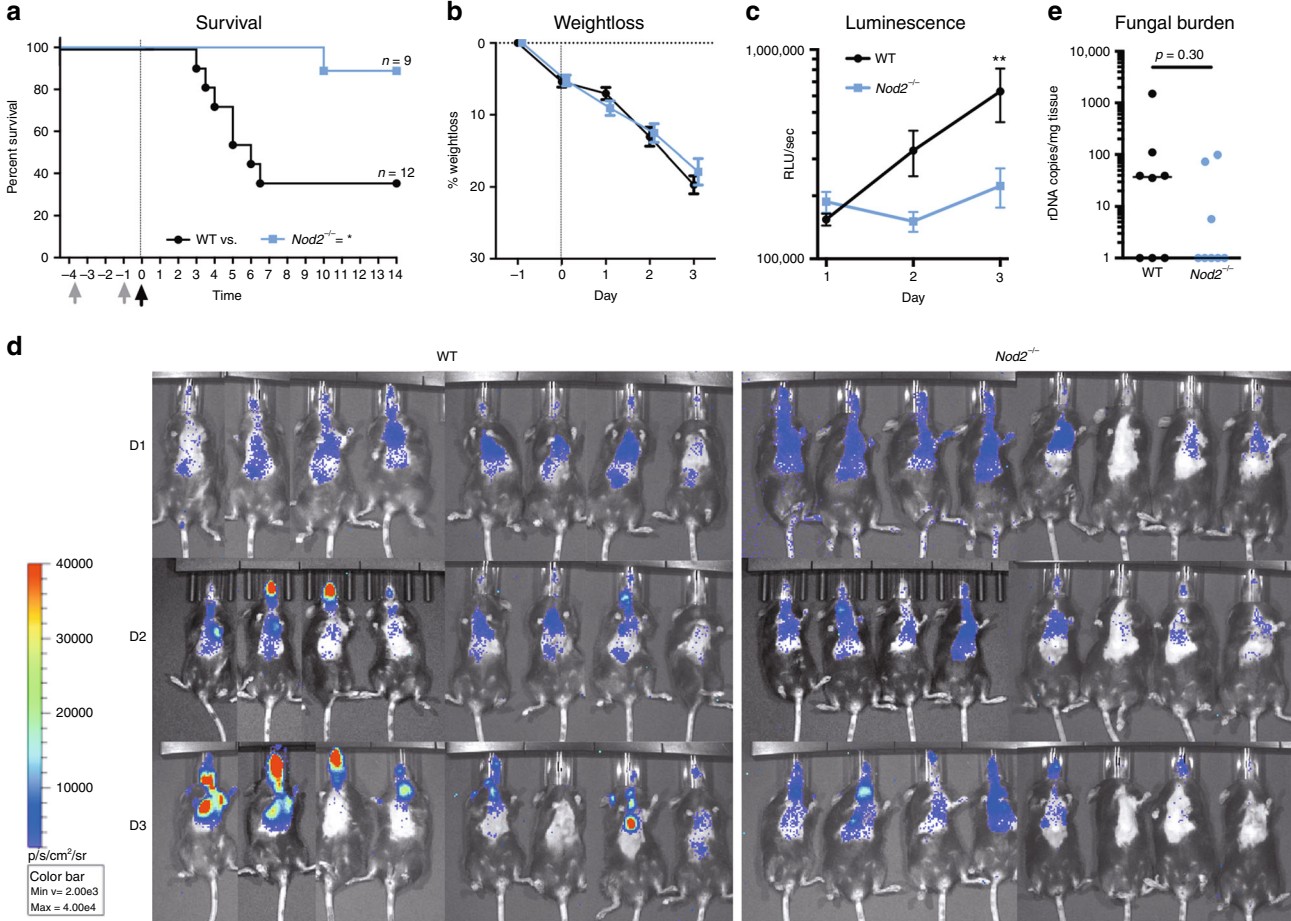

**Fig. 4** $Nod2^{-/-}$ deficient mice have a reduced susceptibility to aspergillosis. Comparison of the susceptibility of wild-type mice (WT; black lines with circles) and $Nod2$-deficient mice ($Nod2^{-/-}$; blue lines with squares) to invasive aspergillosis. **a** Kaplan–Meier survival curve of cyclophosphamide immunosuppressed WT ($n = 12$) and $Nod2^{-/-}$ ($n = 9$) mice infected intranasally with $5 \times 10^4$ conidia. p-values of the Kaplan–Meier curve were determined with the use of the log-rank test. Data represent the cumulative data of four separate experiments. **b** Percentage weight loss following cyclophosphamide immunosuppression and intranasal $Aspergillus$ infection ($5 \times 10^4$/mouse) in WT ($n = 20$) and $Nod2^{-/-}$ ($n = 17$) mice ($p = 0.3515$) (**c, d**) Luminescence signal at day 1 to 3 post infection from the luminescent $Aspergillus$ originating from lung and sinus regions in WT ($n = 20$) and $Nod2^{-/-}$ ($n = 17$) mice. Curves were compared by repeated measurements two-way ANOVA. **e** Fungal burden as determined by amplification of $Aspergillus$ ITS2 regions from lung homogenates. Data in graphs are represented as mean ± SEM or in scatterplots with a line indicating the median. The means were compared using the Mann–Whitney U test, p-values of statistical tests are shown within the graphs, luminescence and weight curves were compared for significance using a two-way repeated measurements ANOVA

the absence of NOD2 positively influences fungal killing we hypothesized that NOD2 activation might impair antifungal host defence. Human MDMs were, therefore, pre-exposed to the NOD2 agonist MDP and subsequently fungal killing capacity was examined. In line with the observation that NOD2 deficiency and silencing is associated with increased $Aspergillus$ killing capacity, NOD2 activation conversely reduced the capacity of human MDMs to kill $Aspergillus$ spores (Fig. 8c).

Several antifungal mechanisms could account for the observed increased killing capacity of $Aspergillus$ in monocytes and macrophages. Phagocytosis and ROS production are well-established factors that influence the fungal killing capacity. Therefore, these two possible mechanisms were systematically addressed to explain increased killing. $Nod2^{-/-}$ BMDMs demonstrated an enhanced capacity to engulf FITC-labelled $A.$ $fumigatus$ conidia, illustrated by a higher percentage FITC-positive macrophages and an overall higher mean fluorescence intensity (MFI) of the macrophages (Fig. 8d), indicating that more conidia were engulfed and more cells were actively engulfing conidia. Similarly, human MDMs in which $NOD2$ was silenced showed a trend towards an increased phagocytosis

(Fig. 8e). Conversely, MDP-stimulated MDMs demonstrate a reduced phagocytosis of FITC-labelled conidia (Fig. 8f).

Although no influence of human NOD2 deficiency on ROS production was found (Fig. 3e), we wanted to validate that ROS production was indeed not influenced by NOD2 deficiency and NOD2 stimulation. BMDMs of WT and $Nod2^{-/-}$ mice stimulated with zymosan demonstrated a similar capacity to produce ROS (Fig. 8g). NOD2 stimulation of human MDMs also did not influence ROS production in response to zymosan stimulation (Fig. 8h). These data suggest that the observed increased killing in the setting of NOD2 deficiency is due to enhanced phagocytosis and not via increased ROS production in contrast to NOD1 deficiency[17].

**NOD2 negatively regulates dectin-1 expression**. One of the most crucial receptors for $A.$ $fumigatus$ recognition and engulfment is dectin-1. Therefore, we investigated whether NOD2 influenced the expression of dectin-1. $Nod2^{-/-}$ BMDMs showed an increased expression of $Clec7a$, the gene encoding dectin-1 (Fig. 8i). Similarly, silencing $NOD2$ in human MDMs slightly

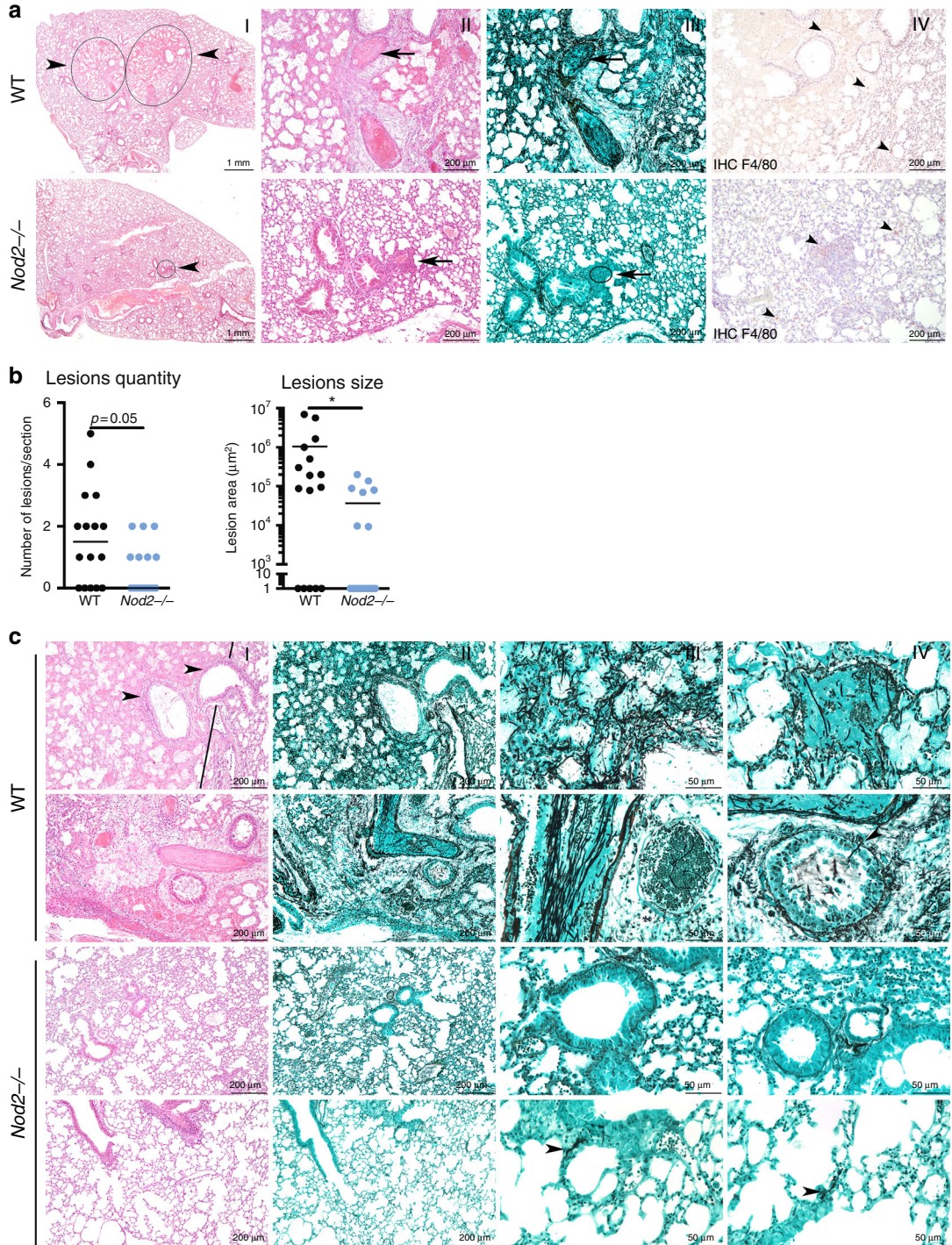

**Fig. 5** $Nod2^{-/-}$ mice show reduced histological damage and fungal burden following *Aspergillus* infection. **a** Histology of lung sections of wild-type and $Nod2^{-/-}$ mice at day 3 pi, stained in HE (I, II), Grocott's Methenamine Silver (III) or labelled using anti-F4/80 antibody (specific for macrophages), counterstained with Haematoxylin staining. Scale bars represent 1 mm (I) and 200 μm (II–IV). **b** Morphometric analysis of the lesions in the whole lung sections using Image J software to quantify the lesions in number and size. **c** Representative lung sections of two additional WT and $Nod2^{-/-}$ mice, stained in HE (I) and Grocott's Methenamine Silver (II–IV). Scale bars represent 200 μm (I, II) and 50 μm (III, IV), means were compared for significance using the Mann-Whitney *U* test

enhanced *CLEC7A* mRNA expression (Fig. 8j). Conversely, MDP stimulation reduced surface dectin-1 expression on human MDMs (Fig. 8k).

**MDP inhibits antifungal immunity in WT cells**. To verify that MDP did not have off-target effects negatively influencing fungal killing, phagocytosis, and dectin-1 expression, monocytes

of healthy volunteers that were wild type for the investigated *NOD2* SNPs were compared with three NOD2-deficient patients. Similarly, fungal killing was assessed in murine BMDMs. MDP significantly reduced in murine BMDMs (Fig. 9a) and in human monocytes fungal killing (Fig. 9b), phagocytosis (Fig. 9c), and dectin-1 expression (Fig. 9d), whereas in the cells of $Nod2^{-/-}$ mice or NOD2-deficient patients no effect of MDP could be detected.

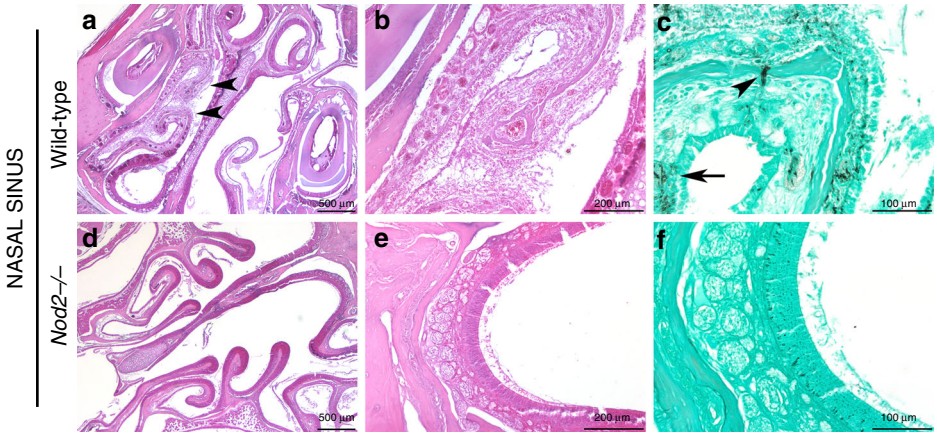

**Fig. 6** histology of nasal sinuses Histology of Nasal Sinuses of (**a**–**c**) wild-type and (**d**–**f**) *Nod2*<sup>−/−</sup> mice at day 3 pi, stained by HE staining at (**a**, **d**) ×2 and (**b**, **e**) ×10 magnification and (**c**, **f**) Grocott's Methenamine Silver staining at ×20 magnification. Scale bars represent (**a**, **d**) 500 μm, (**b**, **e**) 200 μm, and (**c**, **f**) 100 μm

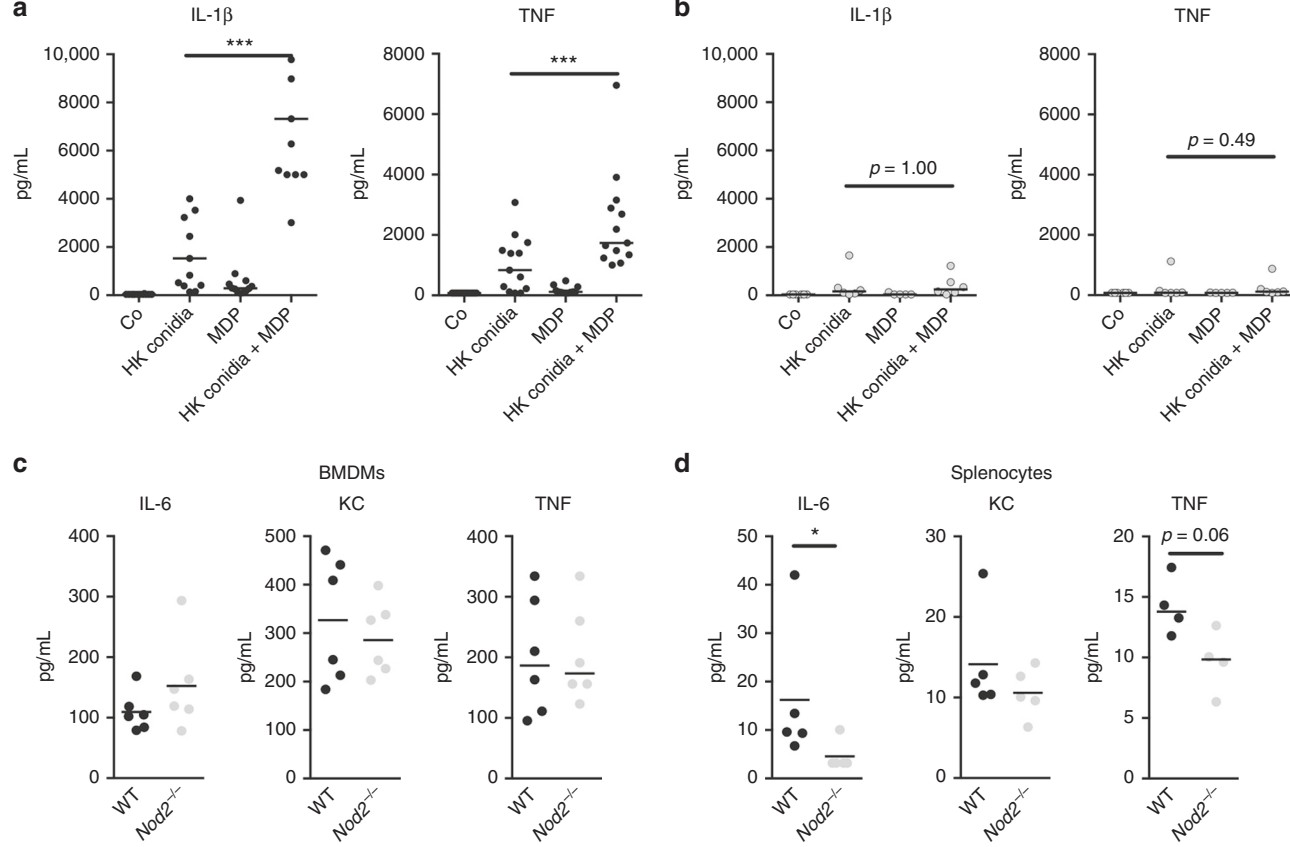

**Fig. 7** NOD2 activation positively regulates cytokine production. **a**, **b** *Aspergillus*-induced IL-1β and TNF levels in the culture supernatants of human PBMCs (5 × 10⁵) of (**a**) healthy volunteers (*n* = 13) represented as black dots or (**b**) NOD2-deficient patients (*n* = 6) represented as grey dots in the presence or absence of (10 μg/mL) MDP. Data is represented as scatter dot plot with median and means were compared using the Wilcoxon signed rank test was paired comparisons. **c**, **d** *Aspergillus*-induced IL-6, KC, and TNF levels in the culture supernatants of murine (**c**) BMDMs and (**d**) splenocytes of wild-type (WT, black dots) and *Nod2*-deficient (*Nod2*<sup>−/−</sup>, grey dots) mice. Data are represented scatterplots with a line indicating the median and means were compared using the Mann–Whitney *U* test

## Discussion

PRRs are key players in activating the antifungal host response during invasive aspergillosis (IA) by inducing cytokine responses and facilitating phagocytosis with subsequent fungal killing. PRRs on the cell surface, such as Toll-like receptors and C-type lectin receptors, have been extensively described in inducing these responses in host defence against *Aspergillus*[31]. Genetic variation

in PRRs is common in the general population, however, in hematopoietic stem cell transplant patients (HSCT), such variations can drastically impact susceptibility to IA[6]. The only intracellular PRRs explored to date, NLRP3 and NOD1 belonging to the NLRs, provide evidence that this class of receptors can modulate host responses against *A. fumigatus*[7,17]. However, one of the most well known NLRs that is directly linked with

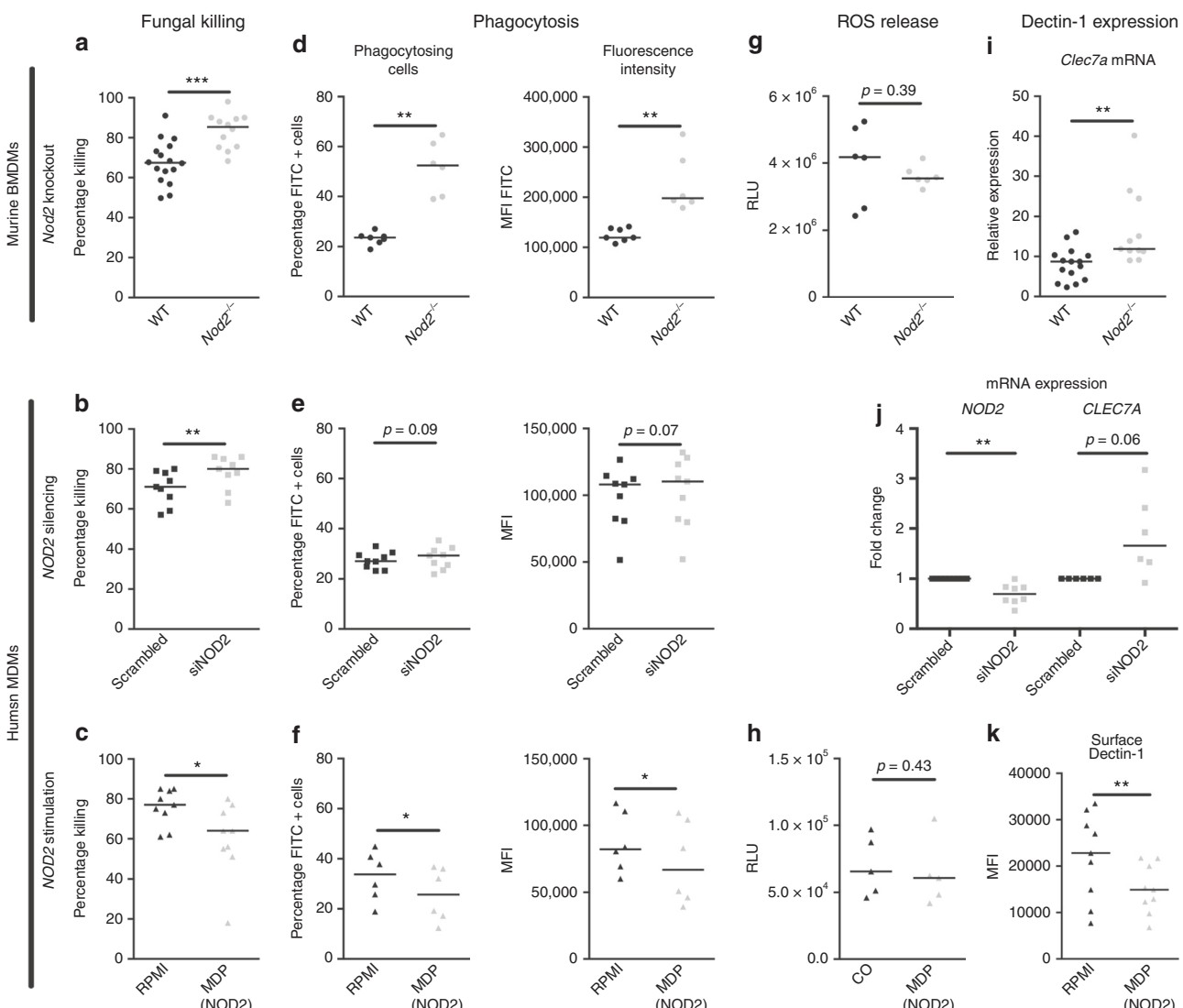

**Fig. 8** NOD2 negatively regulates, fungal killing, phagocytosis, and dectin-1 expression. **a–c** The fungal killing capacity of macrophages ($1 \times 10^5$) assessed by CFU remaining of *A. fumigatus* ($2 \times 10^6$) following exposure for 24 h, in (**a**) wild-type and *Nod2*$^{-/-}$ BMDMs (n = 16 WT, n = 12 *Nod2*$^{-/-}$). **b** human GM-CSF differentiated MDMs treated (n = 9) for 48 h with siRNA targeting *NOD2* or non-targeting siRNA, and **c** human GM-CSF differentiated MDMs (n = 9) treated for 48 h with the NOD2 ligand (10 μg/mL) MDP. Phagocytosis efficiency assessed as percentage of macrophages that engulfed FITC-labelled *A. fumigatus* conidia and mean fluorescence intensity of the total macrophage population in (**d**) wild-type and *Nod2*$^{-/-}$ BMDMs (n = 16 wt, n = 12 *Nod2*$^{-/-}$), (**e**) human GM-CSF differentiated MDMs (n = 9) treated for 48 h with siRNA targeting *NOD2* or non-targeting siRNA, and (**f**) human GM-CSF differentiated MDMs (n = 6) treated for 24 h with the NOD2 ligand (10 μg/mL) MDP. **g, h** The area under the curve of the reactive oxygen species release of (**g**) wild-type (n = 6) and *Nod2*$^{-/-}$ (n = 6) BMDMs or (**h**) human MDMs (n = 5) treated for 24 h with the NOD2 ligand (10 μg/mL) MDP in response to zymosan (150 μg/mL) measured by luminescence signal from luminol conversion over 1 h. **i** Dectin-1 (Clec7a) expression assessed by qPCR in wild-type and *Nod2*$^{-/-}$ BMDMs (n = 14 wt, n = 10 *Nod2*$^{-/-}$), (**j**) *NOD2* (n = 8) *and CLEC7A* (n = 6) mRNA expression in human GM-CSF differentiated MDMs treated for 48 h with siRNA targeting *NOD2* or non-targeting siRNA, and (**k**) Surface dectin-1 expression measured by flow cytometry on human GM-CSF differentiated MDMs treated for 28 h with the NOD2 ligand (10 μg/mL) MDP. Data is represented as scatter dot plot with median with (**a, d, g**) black dots representing wild-type mice and grey dots representing *Nod2* deficient mice, (**b, e, j**) black squares representing human macrophages treated with scrambled siRNA and grey squares human macrophages treated with *NOD2* targeting siRNA, and black triangles representing MDMs without MDP pre-treatment and grey triangles MDMs with MDP pre-treatment. Means were compared using the Mann–Whitney *U* test for murine BMDMs (**a, d, g, i**) and the Wilcoxon signed-rank test was paired comparisons following siRNA treatment (**b, e, j**), or MDP stimulation (**c, f, h, k**)

immunodysregulation that leads to disease[23], namely NOD2, remains largely unexplored in the context of anti-*Aspergillus* host defence[13–16]. Here, we systematically addressed the role of NOD2 in susceptibility to *Aspergillus* infection.

We report an association between *NOD2* genetic variation, *Nod2* deficiency and decreased susceptibility to IA. Specifically, the TT-genotype at P268S confers resistance to IA after HSCT, a finding highlighting a potential NOD2-dependent detrimental

effect on antifungal immunity. A potential limitation of our study is the lack of association for other NOD2 polymorphisms eventually with more noticeable loss-of-function phenotypes. This may, however, be explained by the low allele frequency of such variants, which do not allow accurate risk estimations. A previous study also investigated *NOD2* polymorphisms in association with aspergillosis in HSCT patients. Although in this study a lack of association due to the low frequency of the variants was observed

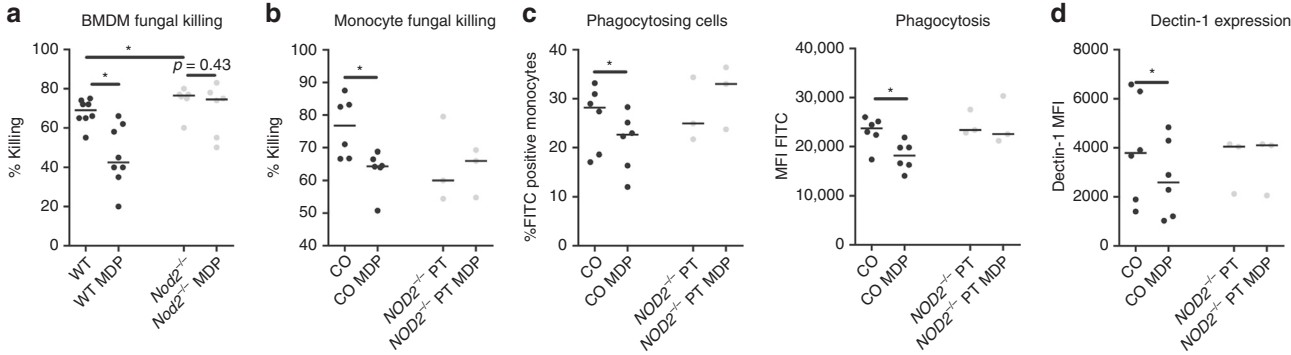

**Fig. 9** MDP negatively affects antifungal host response only in the presence of functional NOD2. **a**, **b** The fungal killing capacity of (**a**) murine BMDMs ($n = 8$ wt and $n = 6$ $Nod2^{-/-}$) and (**b**) human monocytes ($1 \times 10^5$) assessed by CFU remaining of *A. fumigatus* ($2 \times 10^6$) following exposure for 24 h. **c** Phagocytosis efficiency assessed as percentage of monocytes that engulfed FITC-labelled *A. fumigatus* conidia and mean fluorescence intensity of the total macrophage population, (**d**) surface dectin-1 expression measured by flow cytometry in cells of healthy controls (co) ($n = 6$) as well as NOD2-deficient patients ($NOD2^{-/-}$ PT) ($n = 3$). Data are represented as scatterplot with median, black dots represent wild-type and grey dots represent NOD2-deficient cells. Means were compared using the Wilcoxon signed-rank test for paired comparisons except for the comparison of WT with $Nod2^{-/-}$ cells which were compared using the Mann–Whitney $U$ test

for 1007finsC and G908R polymorphisms[32], a strong trend ($p = 0.05$) towards a reduced presence of the mutated R702W allele was observed in IA patients[32].

Functionally, we demonstrate that in particular the 1007finsC polymorphism impacts the response of primary immune cells to *Aspergillus*, namely in cytokine signalling and fungal killing, whereas we only observed an effect of the P268S polymorphism on cytokine responses.

Immunosuppression and cytostatic drugs needed for the treatment of cancer and autoimmune disorders makes patients highly susceptible to invasive fungal infections such as IA. Cyclophosphamide is a drug used to treat hematological malignancies or to suppress the immune system to preventgraft rejection and renders mice highly susceptible to develop infections with *Aspergillus fumigatus*. This immunosuppression allows a low dose of intranasally administered conidia to cause an invasive infection that is lethal within days, which in immunocompetent mice would have been efficiently cleared[30]. *Nod2*-deficient mice were resistant against aspergillosis despite being immunosuppressed and showing severe symptoms of aspergillosis such as weight loss, hunching, head tilting, and circling[30]. The protection observed in $Nod2^{-/-}$ mice was associated with reduced fungal burden and reduced histopathological damage to the lungs. A deficiency in PRRs being protective against lethal aspergillosis is a striking observation. Especially since it is challenging to protect immunosuppressed mice even with available potent antifungal therapies, which often requires combinational therapies to achieve survival of these mice[33,34].

The fact that we observe protection of *Nod2* deficiency in an immunocompromised mouse model raises the question which cells are responsible for the protection. We observed resident macrophages, that remain in the lung even after immunosuppressive therapy, which could potentially mediate fungal killing. In the HSCT patients, the *NOD2* P268S polymorphism was only associated with a reduced incidence of aspergillosis in the donor genotype. The donor genotype will represent the genotype of the patient's myeloid cells following transplantation suggesting that the protective effect of *NOD2* genetic variation lies within the myeloid compartment. Furthermore, we observe that NOD2 negatively affects the antifungal capacity of various types of myeloid cells, including murine BMDMs, human MDMs and human monocytes.

Interestingly, *Staphylococcus aureus* pneumonia in *Nod2*-deficient mice was less severe than in wild-type animals due to

reduced pulmonary inflammation[35]. We observed that some cytokines have lower levels in the BAL of aspergillosis patients having the TT-genotype at P268S (rs2066842). This might indicate a less severe infection that may be related to enhanced fungal killing. Nevertheless, we observed that *NOD2* polymorphisms, as well as the complete deficiency of the receptor, were associated with decreased *Aspergillus*-induced pro-inflammatory cytokine responses. NOD2 stimulation augments *Aspergillus*-induced cytokine responses. It has been widely described that excessive inflammation, and in particular IL-17-mediated inflammation, can result in detrimental immunopathology during *Aspergillus* infections in mice[36–39], but this is primarily observed in situations where the immune system is largely functional such as cystic fibrosis, allergic bronchopulmoary aspergillosis, corticosteroid, and fully immunocompetent models. Although reduced cytokine-driven inflammation can contribute to less damage in certain aspergillosis models, a lower capacity to mount early cytokine response is also known to be a primary risk factor for susceptibility[40]. Based on our data, we can only conclude that NOD2 has a potential to modify *Aspergillus*-induced cytokines in vitro, but it needs to be elucidated whether this in any way contributes to the observed protection in HSCT patients with *NOD2* variants and $Nod2^{-/-}$ mice.

What could then be the mechanism of protection in the setting of genetic *NOD2* deficiency? Carriage of the 1007finsC polymorphism correlates with an increased fungal killing capacity. In addition, *Nod2*-deficient mice demonstrated improved fungal clearance compared to WT mice, which was associated with an absence of histological damage and fungal outgrowth within the lungs. In addition *Nod2* deficiency in murine BMDMs or *NOD2* silencing in human MDMs augments fungal killing, whereas NOD2 stimulation by MDP in human MDMs or monocytes suppresses fungal killing. Subsequently, we systematically addressed whether phagocytosis capacity and/or ROS production, which are well-established mechanisms needed for the killing of *Aspergillus*, would be altered in NOD2-deficient cells. We observed no effect on the ROS production. However, observed that NOD2 negatively regulates phagocytosis. Silencing of *NOD2* gene expression slightly enhanced engulfment of *A. fumigatus* conidia, whereas NOD2 stimulation suppressed the phagocytic capacity of human MDMs and monocytes. $Nod2^{-/-}$ BMDMs were more efficient at engulfing *A. fumigatus* conidia than their WT counterparts. This is in line with a previous report showing that *NOD2* polymorphisms improve phagocytosis of the

gram-negative bacterium *Escherichia coli*[41]. However, in NOD2-deficient patients, we did not observe augmented phagocytosis compared to healthy donor cells. At first sight this might argue against a role for NOD2 in phagocytosis, however, it needs to be taken into account that the rate of phagocytosis is already variable between humans. This is most likely explained by a different genetic background and, in the case of our patients, maybe even immunosuppressive medication. Comparing WT and *Nod2*-deficient murine cells and silencing of the same human cells making them NOD2 deficient practically eliminates these donor factors that contribute to variability of phagocytosis. To prove that NOD2 influences phagocytosis in human cells we made use of the following knowledge. NOD2 is a receptor for derivates of bacterial peptidoglycan, such as MDP, which is present in the peptidoglycan of both gram positive and negative bacteria[8,10,11]. When we studied killing and phagocytosis of *Aspergillus* in the presence or absence of MDP we observed that NOD2 stimulation indeed decreases phagocytosis and killing. By performing these experiments in cells isolated form NOD2-deficient patients we show that MDP in these cells did not influence phagocytosis and killing. These data strengthen the conclusion that NOD2 negatively influences phagocytosis and killing of *Aspergillus* and supports the concept that genetic *NOD2* deficiency could confer protection against invasive aspergillosis by an increased capacity of NOD2-deficient cells to control fungal burden in the host.

One of the crucial PRRs for phagocytosis of *A. fumigatus* is the c-type lectin receptor dectin-1[42,43]. On the one hand, when we studied the expression of dectin-1 in the setting of *Nod2* deficiency or silencing we observed that when phagocytosis was increased this correlated with increased dectin-1 expression. On the other hand NOD2 stimulation with MDP decreased surface dectin-1 expression. The observed correlation between increased dectin-1 expression and increased phagocytosis and killing within the setting of NOD2 deficiency, may argue for a role for dectin-1, but does not exclude that other mechanisms are still playing a role in the observed protection.

Although it has previously been shown that other fungi such as *Candida albicans* are not recognized by NOD2[44,45], the fungal cell wall component chitin/chitosan that is present in both *Aspergillus* and *Candida* has been suggested to be a ligand for NLRs[18,46]. Chitosan activates NLRP3 and thereby activates the inflammasome and induces IL-1β production, whereas chitin did not activate NLRP3[46]. Chitin induces IL-10 dependent on TLR9, mannose receptor and NOD2[18]. These data suggest that NOD2 plays a role in the recognition of fungal molecules such as chitin. However, a different study demonstrated that chitin-induced IL-1Ra production in human PBMCs is independent of NOD2[47]. In addition, chitin can synergize with the NOD2 ligand MDP to augment IL-1β and TNF responses[47], similar to our current observation that *Aspergillus* synergizes with MDP stimulation. Although this underlines that chitin and possibly other fungal cell wall molecules synergize with NOD2 signalling to augment cytokine responses, further studies are required to identify the PAMPs in *Aspergillus* that are recognized by NOD2. Moreover, NOD2 may not be directly involved in recognizing *Aspergillus*, but rather coordinate the responses that are induced by other (membrane-bound) PRRs, for example, the orchestration of phagosome composition. NOD2 synergizes with TLR signalling to yield more potent inflammatory responses[48–52]. Selective modulation of signals from PRRs that recognize *Aspergillus* is a possible mechanism by which the NOD receptors regulate the host response to *A. fumigatus*.

Collectively our data highlight a detrimental effect for NOD2 on antifungal host defence against *A. fumigatus*. This places NOD2 in a unique position in anti-*Aspergillus* host defence. It has the capacity to increase phagocytosis and killing in a ROS

independent way. This could provide a rationale for treating patients that are immunosuppressed, either due to primary immunodeficiency such as chronic granulomatous disease that lack ROS or in patients that receive corticosteroids that suppress immune cells to produce ROS. Moreover, the effects of NOD2 deficiency are in sharp contrast with NOD1 deficiency. NOD1-deficient cells show increased cytokine production in response to *Aspergillus*. This might be beneficial, but could also be detrimental during the natural course of aspergillosis. Moreover, the oxidative burst is significantly higher under NOD1 deficient conditions and is decreased by NOD1 stimulation[17], whereas in NOD2 we do not find an association with altered ROS production. NOD1 and NOD2 are closely related and interact with each other, therefore one would expect they behave similar in anti-*Aspergillus* host defence, but here we demonstrate clear different roles for NOD2 than the previous effects described for NOD1[17]. A potential explanation for the different phenotypes observed with NOD1 and NOD2 deficiency is that *Aspergillus* PAMPs may have different affinities for the two different receptors or that different PAMPs bind and/or activate the receptors. Binding of the receptors by different PAMPs could lead to the fact that both receptors compete for the downstream adapter RICK, which was previously proposed to explain differential regulation of inflammation by NOD1 and NOD2 in arthritis[53]. An alternative explanation for the different phenotypes observed with NOD1 and NOD2 is that one, or both, of these receptors, can, in addition to RICK, induces an alternative-signalling cascade. Of note, it has previously been demonstrated that NOD2 can signal through the intracellular adaptor CARD9[54], which has a strong association with antifungal host response[55,56]. It is tempting to speculate that the detrimental effect of NOD2 may be due to sequestering CARD9 from other receptors requiring CARD9 as a signalling adaptor, such as dectin-1[57], dectin-2[58]. Further studies using NOD1/NOD2, RICK, and NOD2/CARD9 knockout mice and co-precipitations would be required to investigate how the molecular pathways of NOD1 and NOD2 intertwine to mediate detrimental effects on the antifungal host response against *Aspergillus*.

NOD2 deficiency mediates protection against *Aspergillus* in mice, and polymorphisms in *NOD2* alter the susceptibility of HSCT patients to develop aspergillosis. These effects are in the context where NOD2 seems to play a role in the induction of innate and adaptive cytokine responses against *Aspergillus* in humans. The absence of NOD2, however, strongly correlates with an enhancement of fungal killing and phagocytosis, which is independent of ROS. This makes NOD2 an attractive therapeutic target in the treatment of invasive aspergillosis.

## Methods

**Study design**. A total of 310 consecutive haematological patients of European descent undergoing allogeneic HSCT at Instituto Português de Oncologia, Porto, and at the Hospital de Santa Maria, Lisbon, between 2010 and 2014, and respective donors, were included in the genetic association study. The demographic and clinical characteristics of the patients were as previously described[26] and are presented in Supplementary Table 2. Exclusion criteria were the development of fungal infection other than that caused by *Aspergillus spp.*, and pre-transplant fungal infection. It should be noted that this cohort was previously successfully used for identification of genes conferring increased susceptibility to aspergillosis[26]. The sample size was estimated to provide a power of 80% ($1 - \beta = 0.80$) with a type I error below 5% ($\alpha = 0.05$) for genetic variants with minor allele frequencies between 10 and 20% conferring a relative risk of 2.0.

For the functional genomics study, similarly, the cohort was previously successfully used for identification of polymorphisms that lead to reduced cytokine responses[59]. The functional genomics cohort consisted of 200 healthy volunteers, of which approximately 80 (variable per genotype) were included in the current study. Individuals of which the genotype could not reliably be determined using SNP assays, and individuals that were not assessed for cytokine production were excluded from the analysis. The sample size of 80 healthy individuals was estimated to provide a power of 70% ($1 - \beta = 0.70$) with a type I error below 5% ($\alpha = 0.05$) for genetic variants with a minor allele frequencies of 20% conferring an odds ratio

of 2.5. No patients were excluded in these studies. All cytokine and killing assays were performed by a researcher blinded for the genotype.

**Ethics statement**. For the genetic association study, approval was obtained from the Ethics Subcommittee for Life and Health Sciences of the University of Minho, Portugal (125/014 and 014/015), the Ethics Committee for Health of the Instituto Português de Oncologia—Porto, Portugal (26/015), the Ethics Committee of the Lisbon Academic Medical Center, Portugal (632/014), and the National Commission for the Protection of Data, Portugal (1950/015).

For the functional genomics study and patient studies, drawing of blood samples from patients and healthy volunteers was approved by the local ethical board at the Radboud University Nijmegen (Arnhem-Nijmegen Medical Ethical Committee).

For assessment of BAL cytokine levels approval was obtained from the Ethics Subcommittee for Life and Health Sciences of the University of Minho, Portugal (126/014), and the Ethics Committee of the University Hospitals of Leuven, Belgium.

All patients and healthy volunteers provided written informed consent.

Mice were cared for in accordance with Institut Pasteur guidelines, in compliance with European animal welfare regulation. This study was approved by the ethical committee for animal experimentation CETEA (Comité d'éthique en experimentation animale, Project license number 2013-0020). Animal studies were conducted under protocols approved by St. Jude Children's Research Hospital Committee on Use and Care of Animals (protocol no 482-100265-1-/13).

**Healthy controls and *NOD2* deficient patients**. Venous blood samples from healthy controls and patients were obtained and were analysed for polymorphisms in *NOD2* gene (P268S rs2066842, G908R rs2066845, R702W rs2066844 and 1007finsC, rs2066847). DNA was isolated from whole blood by using the isolation Gentra Pure Gene Blood kit (Qiagen), according to the manufacturer's protocol. Gene fragments were amplified and genotyped using commercially available TaqMan SNP Genotyping Assays (Applied Biosystems) according to the manufacturer's protocol on the StepOnePlus system (Applied Biosystems). Quality control was performed by the incorporation of positive and negative controls and duplication of random samples across different plates.

Nine patients with Crohn's disease that were homozygous for the 1007finsC polymorphism were included for studying NOD2 deficiency. Most patients received anti-inflammatory therapy for treatment of their Crohn's disease; Patient 1 Mesalazine 1dd 1000 mg, Patient 2 No immunomodulation, Patient 3 Adalimumab 1 × 2 weeks 40 mg sub cutaneous, Patient 4 mesalazine 3dd 1 g and azathioprine 1dd 200 mg, Patient 5 infliximab every 8 weeks 300 mg iv, Patients 6–9 unknown.

**Genotyping**. DNA was isolated using the Gentra Pure Gene Blood kit (Qiagen), in accordance with the manufacturer's protocol. Genotyping was performed using KASPar assays (LGC Genomics, Hertfordshire, UK) in an Applied Biosystems 7500 Fast Real-Time PCR system for the patient cohort. Mean call rate was >97% for all genotyped SNPs. Quality control for the genotyping results was achieved with negative controls, common and rare homozygous controls (whenever available), and retesting of samples with indeterminate results. Details of the MAF of the polymorphisms in our cohort are provided in Table 1 and linkage disequilibrium for all genotyped SNPs is shown in Supplementary Table 1.

**Aspergillus fumigatus strains**. A clinical isolate of *Aspergillus fumigatus* V05–27 was used for all ex vivo and in vitro stimulations[60]. *Aspergillus* was grown for 7 days at 37 °C on Sabouraud dextrose agar slants poured in T150 cell culture flasks (Corning). Abundant conidia were produced under these conditions. To harvest conidia phosphate-buffered saline (PBS) with 0.05% Tween 80 was poured on the slants and the surface was gently scraped using a cell scraper. To remove hyphae and debris, the conidial suspension was filtered through four layers of sterile gauze. Conidia were counted using a Bürker counting chamber, stored at −20 °C or heat inactivated for 30 min at 90 °C. A concentration of $1 \times 10^7$/mL was used in the experiments unless otherwise indicated. To obtain hyphal fragments, a suspension of $1 \times 10^7$/mL conidia was made in RPMI1640. After 10 h of incubation at 37 °C, the tubes were centrifuged at $1550 \times g$ for 10 min, and the pellet, containing almost exclusively hyphae, was washed twice in PBS and heat inactivated for 30 min at 90 °C.

Heat-inactivated *Aspergillus* conidia ($1 \times 10^7$/mL) were FITC-labelled by incubation with FITC at a final concentration of 0.1 mg/mL (SIGMA) in 0.05 M Na carbonate buffer (pH 10.2) at 37 °C for 1 h. Unbound FITC was washed away by centrifugation three times in PBS–0.1% Tween 20, and labelled conidia were resuspended in RPMI1640, counted and adjusted to a concentration of ($4 \times 10^8$/mL)[61]. For in vivo experiments the luciferase-expressing *Aspergillus fumigatus* 2/7/1 strain was used[29], this strain contains two genomic $luc_{Opt}$ integrations under control of the *Aspergillus gpdA* promotor that regulates stable luciferase expression[62]. The strain has a similar antifungal susceptibility as its parental CBS144.85 strain and demonstrates no growth defects under various in vitro cultivation conditions such as different temperatures and carbon sources[29]. In

corticosteroid immunosuppressed mouse models of aspergillosis[62], the 2/7/1 strain demonstrated a similar virulence as observed for its parental strain CBS144.85[63,64].

**PBMC isolation and stimulation**. Venous blood was drawn into 10 mL EDTA tubes. Blood was diluted in PBS (1:1) and fractions were separated by Ficoll (Ficoll-Paque Plus, GE healthcare, Zeist, The Netherlands) density gradient centrifugation according to the protocol supplied by the manufacturer. Cells were washed twice with PBS and resuspended in RPMI-1640+ (RPMI1640 Dutch modification supplemented with 50 μg/mL gentamicin, 2 mM L-glutamine and 1 mM pyruvate; Gibco, Invitrogen, Breda, The Netherlands).

PBMCs were plated in 96-well round-bottom plates (Corning, NY, USA) at a final concentration of $2.5 \times 10^6$ cells/mL and in a total volume of 200 μL. The individuals in the functional genomic cohort were stimulated with medium (negative control) or live *Aspergillus* at a final concentration of $1 \times 10^7$/mL for 24 h or HI *A. fumigatus* conidia for 7 days (to prevent outgrowth of the fungus influencing the results). The NOD2-deficient patients were stimulated in the presence of 10% serum with the culture medium, live *A. fumigatus* conidia ($1 \times 10^7$/mL), HI conidia ($1 \times 10^7$/mL) or HI hyphae (derived from $1 \times 10^7$/mL conidia). PBMCs in co-stimulation experiments were exposed to 10 μg/mL MDP and subsequently stimulated with medium, HI conidia ($1 \times 10^7$/mL). After stimulation culture supernatants were collected and stored at −20 °C until cytokine measurement.

**Flow cytometry**. Flow cytometry for *Aspergillus*-induced IL-17A+, IL-22+, and IFNγ+ T-cells was performed as described previously[65]. Following 7 day *Aspergillus*-stimulations, culture supernatants were removed and PBMCs were restimulated 4–6 h with PMA (50 ng/mL; Sigma-Aldrich), ionomycin (1 mg/mL; Sigma-Aldrich), and GolgiPlug (BD Biosciences) according to the protocols supplied by the manufacturers. Cells were stained extracellular in a total volume of 50 μL using PE-Cy7–conjugated anti-CD4 monoclonal antibody (eBiosciences, clone RM4–5, dilution 1:20). Subsequently, the cells were fixed and permeabilized with Cytofix/Cytoperm solution (eBioscience) according to the protocol supplied by the manufacturer. Following permeabilization the cells were stained intracellularly with Alexa 647-conjugated anti-IL-17A monoclonal antibody (BD Biosciences, Clone TC11-18H10, dilution 1:6), PE-conjugated anti-IL-22 monoclonal antibody (R&D Systems, Clone 142928, dilution 1:12), and FITC-conjugated anti-IFNγ monoclonal antibody (eBioscience, clone 4s.B3, dilution 1:300) according to the protocols supplied by the manufacturer's. The cells were measured on an FC500 flow cytometer (Beckman Coulter) and the data were analysed using CXP analysis software v2.2 (Beckman Coulter).

The surface dectin-1 expression on human MDMs was assessed following stimulation of with MDP as described above. MDMs were stained in a final volume of 50 μL with FITC-conjugated anti-human CD14 monoclonal antibody (BD; clone TUK 4, dilution 1:20), KromeOrange-conjugated anti-human CD45 monoclonal antibody (Beckman Coulter; clone J.33, dilution 1:10) and APC-conjugated anti-human dectin-1 monoclonal antibody (R&D, clone 259931, dilution 1:10). CD14+ cells were gated within the population of CD45+ cells and subsequently, the Mean fluorescence intensity (MFI) of Dectin-1 was assessed on the CD14+/CD45+ cells (Supplementary Figure 4). The cells were measured on a Cytoflex flow cytometer (Beckman Coulter) and the data were analysed using Kaluza software (Beckman Coulter).

**Aspergillus killing assays**. Freshly isolated PBMCs ($5 \times 10^5$), human GM-CSF monocyte-derived macrophages ($2 \times 10^5$) or murine BMDMs ($2 \times 10^5$) were exposed to *Aspergillus* conidia ($2 \times 10^6$) in 96 well plates a final volume of 200 μL. After 24 h at 37 °C, the cells were washed in water and plated in serial dilution on Sabouraud agar plates. CFUs were counted after 24 h at 37 °C.

**Phagocytosis assays**. Human CD14+ monocytes, human MDMs, or BMDMs were plated in 24 flat bottom plates at $5 \times 10^5$ cells/well. Cells were allowed to phagocytose $5 \times 10^6$ (MOI 1:10) heat inactivated FITC-labelled conidia for 4 h. Subsequently the fluorescence signal of extracellular non-phagocytosed conidia was quenched using 0.2% trypan blue. The cells were measured on a Cytoflex flow cytometer (Beckman Coulter) and the data were analysed using Kaluza software (Beckman Coulter). The monocytes that phagocytosed one or more conidia were enumerated by their positivity for the FITC signal, and could be divided into a FITC negative (monocytes that did not engulf conidia) and a FITC positive (monocytes that engulfed conidia) population. Phagocytosis efficiency was assessed as the mean fluorescence intensity of FITC+ macrophage population (Supplementary Figure 5).

**ROS induction**. The induction of ROS was measured by oxidation luminol (5-amino-2,3,dihydro-1,4-phtalazinedione). PBMCs ($5 \times 10^5$), murine BMDMs ($1 \times 10^5$), human MDMs ($1 \times 10^5$) were resuspended in HBSS and put in dark 96-well plates. Cells were exposed to HBSS, *A. fumigatus* germs ($1 \times 10^7$/mL; PBMCs only) or Zymosan (150 μg/mL). Immediately 20 μL of 1 mM luminol was added. Chemiluminescence was measured in BioTek Synergy HTreader at 37 °C for every min during 1 h.

**Quantitative reverse transcriptase PCR for *CLEC7A* expression**. RNA was isolated according to the protocol supplied with the TRIzol reagent. Isolated mRNA (1 μg) was reverse transcribed into cDNA using the iScript cDNA synthesis kit (BIORAD). Quantitative real-time PCR (qPCR) was performed using Power SYBR Green PCR master mix (Applied Biosystems) and following primers (all manufactured by Biolegio) for human samples hNOD2 Fwd 5′-CCCTGCAGC TGGACTACAACT-3′ and Rev 5′-AGATGCCTCGGTCTGAGATATTG-3′, hGAPDH Fwd 5′-AGGGGAGATTCAGTGTGGGTG-3′ and Rev 5′-CGACCACTT TGTCAAGCTCA-3′ hCLEC7A Fwd 5′-ACAATGCTGGCAACTGGGCT-3′ and Rev 5′-GCCGAGAAAGGCCTATCCAAAA-3′ and the following primer sets form mouse samples mClec7a Fwd 5′-AGGTTTTTCTCAGCCTTGC CTTC-3′ and Rev 5′-GGGAGCAGTGTCTCTTACTTCC-3′, mGapdh Fwd 5′-AG GTCGGTGTGAACGGATTTG-3′ and Rev 5′-TGTAGACCATGTAGTTGAGGT CA-3′. PCR was performed using an Applied Biosystems StepONE PCR system using PCR conditions 2 min 50 °C, 10 min 95 °C followed by 40 cycles at 95 °C for 15 s and 60 °C for 1 min. The RNA genes of interest were corrected for differences in loading concentration using the signal of the housekeeping protein GAPDH.

***NOD2* silencing**. Freshly isolated PBMCs were differentiated to macrophages using 6-day differentiation in 10% human serum supplemented with 5 ng/mL GM-CSF (R&D systems). After differentiation ($1 \times 10^5$) macrophages were seeded in 96 well plates and left for 2 h at 37 °C to subsequently transfect them with 25 nM NOD2 targeting siRNA (on target) or scrambled (non-target) control siRNA (smart pool, Thermo Scientific) for 24 h at 37 °C (Dharmafect, Thermo Scientific). Subsequently, the culture medium was refreshed and cells were used for killing and phagocytosis assays and PCR analysis.

**In vivo experiments**. Mice for in vivo experiments were supplied by the breeding centre R. Janvier (Le Genest Saint-Isle, France). All mice were housed under specific pathogen-free conditions in IVC cages, and fed standard chow and water ad libitum. For the survival experiment in an immunosuppressed background C57BL/6 wild-type (6male/7female), and C57BL/6 *Nod2*$^{-/-}$ (7male/2female) mice (28 to 31 g, 10 weeks old) were used. An estimated power of 80% ($1 - \beta = 0.80$) with a type I error below 5% ($\alpha = 0.05$) for a relative risk of 1.8 was estimated based on a median survival of 4 days in the control group. Mice were separated between genotypes into cages without further randomization and immunosuppressed at day 4 and day 1 before infection by intraperitoneal injection of 200 μL cyclophosphamide (Sigma Aldrich) at 4 mg/mL. At the day of infection, mice were anaesthetized by intramuscular injection (150 μL) of ketamine (10 mg/mL) and xylazine (10 mg/mL) hair was shaved from the ventral lung area and subsequently, mice were inoculated intranasally with $5 \times 10^4$ luciferase expressing *A. fumigatus* 2/7/1 conidia[29] in 25 μL PBS.

In all experiments, survival and weight was monitored in an unblinded fashion during the course of infection.

For histological assessment female C57BL/6 wild-type and C57BL/6 *Nod2*$^{-/-}$ mice (19 to 22 g, 8 weeks old) were used. With 8 mice per group in two separate experiments a power was estimated of 80% ($1 - \beta = 0.80$) with a type I error below 5% ($\alpha = 0.05$) for a variance of 5%. They received similar immunosuppression regimen and were similarly infected as the mice for survival. Weight and bioluminescence were monitored daily during the course of infection. At day 3 the mice were euthanized.

**Bioluminescence imaging**. Bioluminescence imaging was acquired at day 1 post-infection (pi) and was continued on days 2, 3, 4, 6, and 8 pi. Images were acquired using an IVIS 100 system (PerkinElmer) according to the manufacturer's instructions. Analysis and acquisition were performed using Living Image software, version 2.6 (Xenogen). A volume of 100 μL of PBS containing 3.33 mg D-luciferin was injected intraperitoneally before each measurement. During image acquisition, mice were anesthetized using a constant flow of 2.5% isoflurane mixed with oxygen by means of an XGI-8 gas anaesthesia system (Xenogen), which allowed control over the duration of anaesthesia. Images were acquired for 5 min[62]. Quantification of photons per second emitted by each organ was performed by defining regions of interest corresponding to the respective organs of interest (sinus and thorax region), using the Xenogen software Living Image, version 3.0.

***Aspergillus* PCR**. Lung homogenates were obtained following disruption in saline using the Retsch Mixer Mill 301 homogenizer. The fungal burden was determined by amplification of *Aspergillus* ITS2 regions. Briefly, Homogenized tissue samples were used for DNA isolation by using the automated MagNA Pure system and the MagNA Pure LC Total Nucleic Acid Isolation Kit according to manufacturers protocol (Roche Applied Science). PhHV was added to all samples as an internal isolation control. The concentration of total isolated DNA was measured using the Quantus Fluorometer (Promega). *Aspergillus* loads were determined by real-time PCR using the LC480 instrument and the probes master kit (Roche Applied Science). Thermocycling conditions were as follows: 37 °C for 10 min, 95 °C for 10 min, and 50 cycles: 95 °C for 15 s, and 60 °C for 45 s. The rDNA ITS2 region of *Aspergillus fumigatus* was detected by using primers 5′-GCGTCATTGCTGCCC TCAAGC-3′, 5′-ATATGCTTAAGTTCAGCGGGT-3′ and probe Cy5-TCCTCGA GCGTATGGGGCTT-BBQ. The PhHV isolation control was detected by using

primers 5′-GGGCGAATCACAGATTGAATC-3′, 5′-GCGGTTCCAAACGTAC CAA-3′ and probe LC610-TTTTTATGTGTCCGCCACCATCTGGATC-BBQ. For the ITS2 detection, a two-fold dilution series of the cloned PCR product was included to calculate the number of copies per reaction.

**Histology**. Sinuses and lungs were removed and immediately fixed in 10% neutral-buffered formalin. After fixation, sinuses were decalcified for 1 month, using a chelating agent (ethylenediaminetetracetic acid—EDTA) in order to allow routine processing of paraffin while preserving high-quality morphology. Sinus and lung samples were then embedded in paraffin and cut into 4 μm thick sections. Serial sections were stained with haematoxylin and eosin (HE) for assessment histological lesions, Grocott's methenamine silver for fungal detection.

For morphometric analysis, fields at a magnification of ×50, covering the entire lung sections of WT and *Nod2*$^{-/-}$ mice at day 2 post infection were selected and analysed using ImageJ software (http://rsbweb.nih.gov/ij/). We used the software to count the number of lesion foci per lung section, considering ischaemic necrosis foci for wild-type mice and small macrophage infiltrates for *Nod2*$^{-/-}$ mice since ischaemic necrosis was not observed in these mice. Using the software we also measured the size of ischaemic necrosis and macrophage infiltrate foci. Results were expressed as the number and surface of lung lesions, relative to the total lung sections.

To detect the presence of macrophages within the lung tissue, immunohistochemistry analysis using a rat anti-mouse F4/80 monoclonal antibody (AbD Serotec, MCA497G, clone CI:A3-1, diluted 1:400), in sterile phosphate-buffered saline and incubated overnight at 4 °C. The primary antibody was visualized with the N-Histofine rat (Microm) according to the manufacturer's protocol. The colour was developed with 3-Amino-9-EthylCarbazole (AEC chromogen, Sigma). The sections were then counterstained with Meyer's haematoxylin and cover-slipped for microscopic observation.

**Cytokine measurements**. The levels of cytokines in human BAL samples were quantified using the Human Premixed Multi-Analyte Kit (R&D Systems, MN, USA). The cytokine levels in culture supernatants of human PBMCs were measured using commercially available ELISA assays according to the protocol supplied by the manufacturer. IL-1β, TNF, IL-6, IL-17A, and IL-22 assays were from R&D systems and IFNγ was from Sanquin (Amsterdam, The Netherlands). Mouse IL-1β, TNF, IL-6, KC, IL-17A, IL-22, and IFNγ in splenocyte stimulations were measured using the Luminex multiplex platform (Millipore, Billerica, MA).

**Statistical analysis**. For the genetic association study, the probability of IA according to *NOD2* genotypes was determined using the cumulative incidence method and compared using Gray's test[66]. Cumulative incidences of infection at 24 months were computed with the *cmprsk* package for R version 2.10.1[67], with censoring of data at the date of last follow-up visit and relapse and death as competing events. The clinical and genetic variables achieving a *p*-value ≤0.15 in the univariate analysis were entered one by one in a pairwise model together and kept in the final model if they remained significant ($p \le 0.05$). Multivariate analysis was performed using the sub-distribution regression model of Fine and Gray with the *cmprsk* package for R[68].

Data are presented as scatterplots representing individual data points and a line indicating the median value or as graphs ±SEM. Data from functional genomic experiments, in vitro experiments and in vivo experiments was subjected to D'Agostino & Pearson omnibus normality test and was not normally distributed. No samples/animals were excluded from analysis in the in vivo experiments. For in vitro experiments all data points are shown without exclusion. In the functional genomic experiments only healthy individuals whose genotype could not accurately be determined were excluded from the studies. Unless otherwise indicated the Mann-Whitney U test was used to determine statistical significant differences between experimental groups and for paired analysis such as with MDP stimulation or siRNA treatment the paired Wilcoxon signed-rank test was used with *$p < 0.05$, **$p < 0.01$, ***$p < 0.001$, and ****$p < 0.0001$. All data were analysed using GraphPad Prism v6.0.

**Data availability**. The authors declare that the data supporting the findings of this study are available within the paper and its Supplementary Information Files. All relevant data are available by request from the authors, with the restriction of data that would compromise patient confidentiality.

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

## Acknowledgements

We thank C. Wertz and M. Fanton D'Andon for providing *Nod2*-deficient mice, M. Schlotter for organizing patient inclusion, B. Rösler for assistance with flowcytometry. We also thank the NOD2-deficient patients for contributing to our study by providing blood samples. M.S.G. was supported by the Erasmus lifelong learning program. F.L.v.d. V. was supported by the E-rare project EURO-CMC. M.O. was supported by the NWO, 016.176.006). A.C. and C.C. were supported by the Northern Portugal Regional Operational Programme (NORTE 2020), under the Portugal 2020 Partnership Agreement, through the European Regional Development Fund (FEDER) (NORTE-01-0145- FEDER-000013), and the Fundação para a Ciência e Tecnologia (FCT) (IF/00735/2014 to A.C. and SFRH/BPD/96176/2013 to C.C.).

## Author contributions

M.S.G., A.C., O.I.-G., and F.L.v.d.V. conceived and designed the study. M.S.G., C.C., M.J., R.K.S.M., S.M.G., A.A., R.L., M.O., O.R., G.J., C.F., W.M., and O.I.-G. performed experiments. M.S.G., C.C., M.J., R.K.S.M., A.A., G.J., W.M., A.C., O.I.-G., and F.L.v.d.V. analysed the data. D.J.d.J., J.F.L., A.C.J., K.L., and J.M. included patients. T.-D.K. provided valuable reagents and cell lines. M.S.G., M.J., and O.I.-G. wrote the first draft of the manuscript. M.S.G., W.M., T.-D.K., A.C., O.I.-G., and F.L.v.d.V. revised the manuscript.

## Additional information

**Competing interests:** The authors declare no competing interests.

