## [Peer Review File · Nature Communications]

Reviewers' comments:

Reviewer #1 (Fungal infection, host polymorphism)(Remarks to the Author):

This report by Gresnigt et al provides an important advance via defining the SNPs in NOD2 that confer resistance to invasive aspergillosis (IA) after HSCT. The authors have also used a mouse model to show that NOD2 deficiency protects against IA and have attempted to provide mechanistic insight for the role of Nod2 in immune response during IA. Although the report describes a novel role for NOD2 during IA and attempts have been made to understand the role of NOD2 in anti-Aspergillus host defense, the report does not provide a concrete mechanistic insight into how functional deficiency of NOD2 confers protection during IA. That is, it is unclear whether cytokine responses and/or moderation of inflammation and/or direct fungal killing are in play. Below I have outlined the major points which require further attention.

- 1) One possible mechanism as described by the authors, mentioned under Discussion (lines 422-425), is that inflammatory damage may be one contributing factor. Although, the data from Figure 2 for IL1-beta and Figure 3 supports this argument, it is unclear why data for live conidia was not shown as it was shown for other cytokines in Figure 3A.
- 2) In the mouse model, cytokine levels in BAL fluid of the WT and KO mice are not different. These data are in direct contrast with what the authors have concluded on lines 425-427. Moreover, the histology data showed in Figure 5D (panels I and II) do not represent the quantification presented in Figure 5E. The KO histology seems to show increased inflammation. The authors should provide more conclusive data indicating less tissue damage in their KO mice. To that end, they may perform LDH/albumin quantitation in BALF/lung homogenates as surrogates for tissue damage/vascular permeability.
- 3) The authors explain that improved killing of Aspergillus conidia by NOD2 deficient cells ex vivo may also explain the improved outcomes. Despite convincing data, the mechanistic detail of enhanced killing is lacking. Authors postulate that improved phagocytosis can explain the enhanced killing (lines 437-438), however their data in Figure 3F do not support this claim. The authors may wish to analyze phagocytosis using mouse alveolar macrophages, to overcome limiting availability of human monocyte-derived macrophages. Authors may also consider performing transcriptional analysis using RNA-Seq/Nanostring for mouse BMDMs from WT and KO (in presence of Aspergillus conidia). This may aid in identification of possible mechanistic targets that may have a role in enhanced killing.
- 4) In Figure 6B, the authors display that MDP leads to a decreased Aspergillus killing by macrophages from healthy human volunteers. Is it true in case of mouse BMDMs as well? Accordingly, the BMDMs from Nod2^{-/-} mice should also be analyzed for killing in presence of MDP to confirm that MDP does not affect the enhanced killing of the Nod2^{-/-} BMDMs. This will further support a direct role for Nod2 in Aspergillus killing.

- 5) For Figure 2, as per the legend, panels A and C are for stimulations by live conidia, while B & D are for stimulations by HI conidia. It is unclear, why the 3 cytokines were analyzed with live while the other three are analyzed for HI conidia.
- 6) For Figure 3E, plotting the AUC would be more informative.
- 7) Are Figure 5B-C and Figure S3 part of the same experiment? What are the distinctions? Please clarify if the bioluminescence values displayed in Figure 5B, also describes the signal originating from nasal and lung regions? Provide statistics for Figure S3.
- 8) For Figure 5D, the color-balance between WT and KO is significantly different. Provide images with comparable color-balance for ease of interpretation. Also, provide whole lung micrograph as a supplement to better visualize the number of lesions and their area.
- 9) In Figure 5D, the arrowheads for WT (in I) do not correspond to the magnified region on the right in II. Please rectify that. Additionally, provide a description for the arrows and arrowheads in figure legend.
- 10) In Figure 5F, normalize the rDNA copies to the amount of total extracted DNA (i.e. rDNA copies/ μ g of total extracted DNA) for more precise comparison across samples.
- 11) Provide uniform tick-marks for Y-axis of Figure 6A-6B.

Reviewer #2 (Polymorphism, GWAS, innate)(Remarks to the Author):

The authors examine the role of four NOD2 polymorphisms (rs2066842 [P268S]; rs2066844 [R702W]; rs2066845 [G908R]; rs2066847 [1007fs]) on susceptibility to invasive aspergillosis (IA) in a cohort of patients with HSCT (310 recipients and 310 donors). This cohort was previously used to analyze the role of polymorphisms within IL-10 with the same phenotype (published elsewhere). Presence of 1 or 2 copies of the T allele for rs2066842 in HSCT donor was associated with a decreased cumulative incidence of IA (without correction for multiple testing). There was no association between 3 other variants (rs2066844, rs2066845 and rs2066847) with IA, but these SNPs have much lower minor allele frequencies (MAF). The association is supported by functional experiments, showing different (or trends toward different) cytokine production in rs2066842 T carriers (assuming that SNPs rs numbers were inverted in Figure 2) compared to the other individuals. Further experiments support a role for NOD2 deficiency in susceptibility to / protection from IA, which are performed by using samples for NOD2-deficient individuals (i.e. carriers of the 1007fs polymorphisms) or by using NOD2-deficient mice.

Numerous genetic association studies have been reported for susceptibility to invasive aspergillosis over the last decade, some of which have been limited by lack of reproducibility and/or lack of functional evidence supporting the association. In this manuscript, the authors show a potentially interesting association of a NOD2 SNPs (MAF=0.1) with IA. The strength of the study is the importance given to the functional work. However, the SNPs associated with IA is represented in a limited number of experiments (Figure 2A & B), the most striking ones being obtained from individuals carrying another polymorphism (rs20666847) which has a much lower allele frequency (<0.01) and thus could not be associated with the clinical phenotype. Altogether, the observation is interesting but some steps in the association study shall be clarified.

- Among the 4 SNPs selected, 3 have a MAF <0.01. Could the authors provide power calculation to detect association with IA in this cohort ? This power is likely to be very low, so what was the rationale for selecting such very rare SNPs ?
- Did the investigators solely genotype polymorphisms in IL-10 and NOD2 in the clinical cohort of HSCT patients ? If so, what was the rationale for selecting these genes and not other ones ? If not, issues regarding correction for multiple testing should be addressed. This is particularly relevant as there is no replication in another cohort to confirm the genetic association.
- What was the method used for genotyping ? Was it part of a large-scale (or genome-wide scan) or just a few candidate genes ? Could the authors provide the MAF for the selected SNPs in the population studied and values for HWE. Is rs2066842 in linkage disequilibrium with rs20666847 (a LD table for R² and D' might be useful).
- Most patients undergoing HSCT receive mold-active prophylaxis, which may be administered for different duration depending of several factors. Is the association still significant when accounting for such prophylaxis in a time-dependant way ?
- Was the association still significant when patients with infections due to mold other than Aspergillus accounted for ?

Were the Crohn's disease patients providing samples under immunosuppressive treatment, as this might have artificially influenced the analyses.

Reviewer #3 (Aspergillus, mouse infection models)(Remarks to the Author):

An interesting paper in which the role of the NOD2 in host defence against pulmonary aspergillosis is investigated. Interestingly although NOD2 detects peptidoglycan, not present in *Aspergillus*, some data implicates it in *Aspergillus* immunity. NOD2 may however recognise chitin.

The authors first show that SNPs in NOD2 influence probability to IA in HSCT. This was in the donor suggesting a defect in epithelial sensing rather than myeloid sensing. SNP P268S CC increased risk of IA.

Whilst I am not an expert in GWAS studies the data appear to be clear from the figures provided. However I do note the group have shown a lot of associations with a small cohort when compared to other GWAS studies. This could partly be because in immunosuppression there are a number of subclinical polymorphisms that could become relevant thus increasing the number of relevant SNPs, however I would advise a bioinformatician assess this data. For instance is there a correction for multiple testing?

The authors then go on to analyse PBMC responses to *A. fumigatus* in individuals with these SNPs. The authors find that the TT genotype of P268S results in less cytokine production. Therefore it appears that low risk for IA in HSCT for the TT genotype correlated with low cytokine production. The impact on fungal killing was only evident in a frameshift, resulting in increased fungal killing.

The authors then assessed the impact of NOD2 deficient frameshift alleles on *A. fumigatus* responses. These cells had reduced cytokines and reduced CD4 T cells. To correlate findings, NOD2 deficient mice were immunosuppressed with cyclophosphamide then infected with *A. fumigatus*. Under these conditions there was improved survival of NOD2 $-/-$ mice. Bioluminescence demonstrated increased burden in WT mice. The NOD2 $-/-$ mice had less evidence of invasive aspergillosis on histopathology although I would have liked to have seen some objective quantitative analysis of histopathology inflammation with imaging software if possible.

The authors then hypothesize that the reduced susceptibility of NOD2 $-/-$ may be due to increased killing capacity. Nod2 $-/-$ BMDM had increased killing, however the authors do not show why.

The authors then propose that the protective effect of NOD2 dysfunction is probably due to reduced inflammation, however this does not entirely make sense as other inflammatory PRRs confer susceptibility to aspergillosis. I think it is disappointing a mechanistic explanation is not found. The paper is otherwise novel and interesting, and will influence the field.

I was not clear what background the NOD2 $-/-$ mice were in, not completely clear in methods. Otherwise possible to replicate the experiments. Figure S1 has no title.

Replies to Reviewers' comments:

We thank the reviewers for their constructive review of the manuscript. We believe that the corrections and new data, derived from the reviewers' comments and suggested experiments, have significantly improved the manuscript.

Please find below the point-by-point response to the reviewers' concerns in **bold font**. The new experimental data to address these concerns has been included as new Figures, or adapted in supplementary Figures in the revised version of the manuscript. In the revised version of the manuscript, the rewritten text and new text addressing specific issues raised by the reviewers has been highlighted in green.

Point to point replies

Reviewer #1 (Fungal infection, host polymorphism)(Remarks to the Author):

This report by Gresnigt et al provides an important advance via defining the SNPs in NOD2 that confer resistance to invasive aspergillosis (IA) after HSCT. The authors have also used a mouse model to show that NOD2 deficiency protects against IA and have attempted to provide mechanistic insight for the role of Nod2 in immune response during IA. Although the report describes a novel role for NOD2 during IA and attempts have been made to understand the role of NOD2 in anti-*Aspergillus* host defense, the report does not provide a concrete mechanistic insight into how functional deficiency of NOD2 confers protection during IA.

That is, it is unclear whether cytokine responses and/or moderation of inflammation and/or direct fungal killing are in play. Below I have outlined the major points which require further attention.

We appreciate this final comment of the reviewer since we did not have clear data to provide a conclusion whether cytokine responses or killing was the main mechanism. With the comments of the reviewer we have been able to conclude that cytokine responses are different in NOD2 deficiency, but are unlikely to explain the increased susceptibility and that the increased killing due to enhanced phagocytosis in the absence of NOD2 is the main mechanism (see below).

1) One possible mechanism as described by the authors, mentioned under Discussion (lines 422-425), is that inflammatory damage may be one contributing factor. Although, the data from Figure 2 for IL1-beta and Figure 3 supports this argument, it is unclear why data for live conidia was not shown as it was shown for other cytokines in Figure 3A.

Induction of the cytokines IL-17, IL-22 and IFN γ requires an incubation of 7 days for the cytokines to reach detectable levels (see Raijmakers et al Sci Rep. 2017 for dynamics of IL-17). This is due to the fact that these cytokines are primarily derived from T-cells, which require expansion of specific clones for the cytokine production. Since *Aspergillus* would outgrow the human PBMCs in a period of 7 days, we are unable to perform experiments for 7 days with live *A. fumigatus* conidia and are restricted to the use of heat inactivated fungi. We thank the reviewer for this comment since for us it is logical but we should explain this, we have now explained this in the methods (line 669).

2) In the mouse model, cytokine levels in BAL fluid of the WT and KO mice are not different. These data are in direct contrast with what the authors have concluded on lines 425-427.

The reviewer is correct that inflammation in the BAL fluid of WT and KO mice is not different. In lines 439-442 we speculate that there might be reduced inflammatory responses in the absence of NOD2, although the data in BAL from mouse experiments did not show this. We based this speculation on our observation that human PBMCs with NOD2 deficiency or NOD2 SNPs have reduced inflammatory responses (presented in figures 2A-D and 3A,B), and that such a reduced inflammatory response might contribute to the differences in mortality.

We agree with the reviewer that based on the BAL data from the mouse model this speculation of reduced inflammation in the context of *Aspergillus* infection is weakened. We have therefore performed cytokine measurements in the BAL of the patients with aspergillosis. Generally cytokine levels are lower in the BAL of carriers of the TT genotype, with a significant lower level of IL-10 and IL-8 and a trend towards lower level of TNF α . We included these additional data as figure 1C and

elaborated our discussion (lines 443-465) on the influence of inflammation in the observed phenotype in patients/mice with NOD2 SNPs/deficiency.

In addition, we have performed cytokine measurements using splenocytes of WT and *Nod2*^{-/-} and observed a significantly reduced IL-6 production and a trend towards reduced TNF α and KC production, however, using *Nod2*^{-/-} BMDMs we did not observe a significant difference in cytokine release in comparison to WT BMDMs. We included these additional data as figure 6C and D.

The fact that we do not observe a difference in cytokine levels in the BAL of infected mice might be due to the fact that BAL samples were obtained at a suboptimal time point for cytokine assessment. Conversely, it may be that the effects of NOD2 deficiency on cytokine responses is more pronounced in a human setting whereas other influences like the influence of NOD2 on the fungal killing capacity is observed across species. We discussed this issue on lines 454-465, and toned down our conclusion that the observed differences in susceptibility are due to differences in inflammation.

Moreover, the histology data showed in Figure 5D (panels I and II) do not represent the quantification presented in Figure 5E. The KO histology seems to show increased inflammation. The authors should provide more conclusive data indicating less tissue damage in their KO mice. To that end, they may perform LDH/albumin quantitation in BALF/lung homogenates as surrogates for tissue damage/vascular permeability.

We modified the histology figures and all histology data is presented in the new figure 5. We modified panels I (wild-type and *Nod2*^{-/-}), II (*Nod2*^{-/-}), III (wild-type and *Nod2*^{-/-}) (now figure 5A), for a better visualization of the lesions, and we described more precisely the lesions in the results section (lines 266-278). We changed panel I to show a low magnification, where circles delimitate the parenchyma destruction. We observed large area of ischemic necrosis in wild-type animals in contrast to only small inflammatory infiltrates in *Nod2*^{-/-} mice. Panel II now shows a higher magnification of the same lesions, that are definitively more severe for the wild-type group, in agreement with quantification presented in Figure 5B.

We agree with the reviewer that quantitation of LDH and/or albumin will provide more detailed data on tissue damage. However, unfortunately there is no sample left from our experiments for LDH/Albumin quantitation. We do however believe that with improvement of the histology images, and addition of representative histology of two additional WT mice and two *Nod2*^{-/-} mice (new figure 5C) we can definitely conclude that there is less histopathological damage in the lungs of *Nod2*^{-/-} mice.

3) The authors explain that improved killing of *Aspergillus* conidia by NOD2 deficient cells ex vivo may also explain the improved outcomes. Despite convincing data, the mechanistic detail of enhanced killing is lacking. Authors postulate that improved phagocytosis can explain the enhanced killing (lines 437-438), however their data in Figure 3F do not support this claim. The authors may wish to analyze phagocytosis using mouse alveolar macrophages, to overcome limiting availability of human monocyte-derived macrophages.

The reviewer raises an important point that indeed the phagocytosis data from NOD2 deficient individuals does not support the claim that enhanced phagocytosis could be the responsible factor. The problem is that this analysis is underpowered since we only assessed the phagocytic capacity in the last two patients that we included. Furthermore, these assays were performed using PBMCs rather than human monocytes or macrophages, which one could argue to be suboptimal for phagocytosis assessment. In addition, as raised by reviewer 2, these patients were on immunomodulatory therapy for their Crohn's disease, which may have influenced the analysis.

Therefore, as suggested, we assessed the phagocytic capacity in murine macrophages. Due to the difficulty of obtaining enough macrophages we used bonemarrow derived macrophages, rather than alveolar macrophages, in addition we chose this model since our previous killing experiments were performed using this model.

Our results demonstrate that *Nod2*^{-/-} macrophages have an enhanced uptake of FITC labeled *A. fumigatus* conidia, illustrated by the higher number of FITC+ macrophages (more macrophages engulf conidia) as well a higher mean fluorescence intensity of the macrophages (more conidia are engulfed per macrophage). We have included these additional data as figure 7D.

We also wanted to validate that this is not only observed in murine macrophages, but also in human macrophages. Therefore we performed siRNA treatment of human monocyte derived macrophages (MDMs). We observed a trend towards improved phagocytosis when *NOD2* gene expression was silenced in human monocyte derived macrophages. We have included these additional data as figure 7E. Conversely, NOD2 stimulation significantly reduced the capacity of human monocyte derived macrophages to engulf FITC labeled *A. fumigatus* conidia. We have included these additional data as figure 7F.

Authors may also consider performing transcriptional analysis using RNA-Seq/Nanostring for mouse BMDMs from WT and KO (in presence of *Aspergillus* conidia). This may aid in identification of possible mechanistic targets that may have a role in enhanced killing.

When investigating the effect of another NOD receptor (the NOD1 receptor) we observed that NOD1 regulates the induction of reactive oxygen species (Gresnigt et al 2017 *Frontiers Immunology*).

We performed additional experiments to investigate whether NOD2 deficiency in murine BMDMs or NOD2 stimulation in human MDMs modulated oxidative burst. In line with our data in NOD2 deficient patients we observed that unlike NOD1, NOD2 did not modulate oxidative burst, highlighting that the mechanism is different than NOD1. We included these data as figure 7G and H.

However, as described above we were able to pinpoint that NOD2 negatively regulates phagocytosis of *Aspergillus* conidia. Being one of the most crucial receptors for engulfment of phagocytosis, we decided to examine dectin-1 expression. In a pilot experiment we

assessed whether NOD2 silencing, like NOD1 silencing, also enhanced dectin-1 expression, however we were only able to find a trend towards a higher dectin-1 expression (P = 0.06).

In order, to identify a mechanistic target we have performed additional experiments to see whether NOD2 in other cells also influences dectin-1 expression significantly like we previously reported for NOD1. We found that *Nod2*^{-/-} bone marrow derived macrophages have an increased *Clec7A*. Conversely, we found that NOD2 stimulation of human MDMs reduces dectin-1 expression on macrophages. Collectively we included the data on BMDMs, silencing and NOD2 stimulation in figure 7 G, H and I respectively.

Dectin-1 expression

4) In Figure 6B, the authors display that MDP leads to a decreased *Aspergillus* killing by macrophages from healthy human volunteers. Is it true in case of mouse BMDMs as well? Accordingly, the BMDMs from *Nod2*^{-/-} mice should also be analyzed for killing in presence of MDP to confirm that MDP does not affect the enhanced killing of the *Nod2*^{-/-} BMDMs. This will further support a direct role for *Nod2* in *Aspergillus* killing.

The reviewer raises an important point that as a proof of principle, the MDP mediated inhibition of fungal killing should be absent in a *Nod2*^{-/-} background. Therefore we have conducted additional experiments using murine BMDMs and monocytes of NOD2 deficient individuals. We observe that MDP decreases *A. fumigatus* killing in murine BMDMs and in human monocytes. Using the monocytes we also observe reduced phagocytosis of FITC labeled *A. fumigatus* conidia and dectin-1 expression in Healthy controls, but not in patients lacking NOD2. We included these additional data in figure 8

5) For Figure 2, as per the legend, panels A and C are for stimulations by live conidia, while B & D are for stimulations by HI conidia. It is unclear, why the 3 cytokines were analyzed with live while the other three are analyzed for HI conidia.

As explained under point 1 induction of the cytokines IL-17, IL-22 and IFN γ requires an incubation of 7 days for the cytokines to reach detectable levels. *Aspergillus* would outgrow the human PBMCs in a period of 7 days, therefore we use heat-inactivated conidia in stead.

For the cytokines IL-1 β , TNF α and IL-6 24 hours incubation is enough to get detectable levels of these cytokines. As heat inactivated conidia tend to induce a bit lower cytokine levels (See figure 3A) we decided to use live conidia for these assays.

6) For Figure 3E, plotting the AUC would be more informative.

In the revised figure we plotted the AUCs instead of the X-Y plots as suggested.

7) Are Figure 5B-C and Figure S3 part of the same experiment? What are the distinctions? Please clarify if the bioluminescence values displayed in Figure 5B, also describes the signal originating from nasal and lung regions? Provide statistics for Figure S3.

Figure 5B and C are the luminescence data of the experiments where mice were sacrificed at day 3 post infection for assessment of the histology etc. Figure S3 is the luminescence data from the mice that were used for the survival studies. This is also the reason why WT mice are dropping out at the later time points. The quantification of the luminescence for figure S3 and figure 5 was performed in the same way. We have adapted the figure legend from figure 5 to clarify that the signal was originating from the nasal and lung regions. In addition we have provided statistics for figure S3, which show a strong trend towards a reduced fungal burden at day 2 pi ($p = 0.07$).

For increased power of the fungal burden analysis we combined the luminescence data of the experiment where mice were sacrificed for histological assessment and the luminescence signal from the first 3 days of the survival experiment (Figure 4C) we observe that the fungal derived luminescence signal significantly higher in WT mice compared to *Nod2*^{-/-} mice. We also combined the weight loss for the first 3 days in Figure 4B for an increased clarity and avoiding the reporting of same results twice.

8) For Figure 5D, the color-balance between WT and KO is significantly different. Provide images with comparable color-balance for ease of interpretation. Also, provide whole lung micrograph as a supplement to better visualize the number of lesions and their area.

We thank the reviewer for this comment. We have accordingly adjusted the color balance of the figures for a better comparison and interpretation of the histology between WT and *Nod2* knockout. We have additionally changed panel I (figure 5A) to show a lower magnification (x2; lower magnification of our microscope), in order to allow a better visualization of the lesions. In addition, we have included histology images from two additional wild type mice and two *Nod2*^{-/-} ko mice for a more detailed overview of the mice in terms of histopathological damage (Figure 5C).

9) In Figure 5D, the arrowheads for WT (in I) do not correspond to the magnified region on the right in II. Please rectify that. Additionally, provide a description for the arrows and arrowheads in figure legend.

In order to make sure that the magnified regions correspond with the arrowheads, we have modified the panel I with a lower magnification (as requested under point 8). Using arrowheads we highlight the localization of the lesions, which are the magnified region on panel II. In addition we also modified results section as requested to provide a detailed description for the arrowheads and regions observed in the histology (Lines 267-294).

10) In Figure 5F, normalize the rDNA copies to the amount of total extracted DNA (i.e. rDNA copies/ μ g of total extracted DNA) for more precise comparison across samples.

We normalized the rDNA copies to the amount of tissue that was used for DNA extraction, since we unfortunately did not have data on total extracted DNA. In addition we log transformed the axis to be consistent with previous papers that used this method for fungal burden (Arendrup 2010 PLoS ONE).

11) Provide uniform tick-marks for Y-axis of Figure 6A-6B.

We have changed the tick-marks on the Y-axis in these figures

Reviewer #2 (Polymorphism, GWAS, innate)(Remarks to the Author):

The authors examine the role of four NOD2 polymorphisms (rs2066842 [P268S]; rs2066844 [R702W]; rs2066845 [G908R]; rs20666847 [1007fs]) on susceptibility to invasive aspergillosis (IA) in a cohort of patients with HSCT (310 recipients and 310 donors). This cohort was previously used to analyze the role of polymorphisms within IL-10 with the same phenotype (published elsewhere). Presence of 1 or 2 copies of the T allele for rs2066842 in HSCT donor was associated with a decreased cumulative incidence of IA (without correction for multiple testing). There was no association between 3 other variants (rs2066844, rs2066845 and rs20666847) with IA, but these SNPs have much lower minor allele frequencies (MAF). The association is supported by functional experiments, showing different (or trends toward different) cytokine production in rs2066842 T carriers (assuming that SNPs rs numbers were inverted in Figure 2) compared to the other individuals. Further experiments support a role for NOD2 deficiency in susceptibility to / protection from IA, which are performed by using samples for NOD2-deficient individuals (i.e. carriers of the 1007fs polymorphisms) or by using NOD2-deficient mice.

Numerous genetic association studies have been reported for susceptibility to invasive aspergillosis over the last decade, some of which have been limited by lack of reproducibility and/or lack of functional evidence supporting the association. In this manuscript, the authors show a potentially interesting association of a NOD2 SNPs (MAF=0.1) with IA. The strength of the study is the importance given to the functional work. However, the SNPs associated with IA is represented in a limited number of experiments (Figure 2A & B), the most striking ones being obtained from individuals carrying another polymorphism (rs20666847) which has a much lower allele frequency (<0.01) and thus could not be associated with the clinical phenotype. Altogether, the observation is interesting but some steps in the association study shall be clarified.

- Among the 4 SNPs selected, 3 have a MAF <0.01. Could the authors provide power calculation to detect association with IA in this cohort ? This power is likely to be very low, so what was the rationale for selecting such very rare SNPs ?

The reviewer is correct in pointing out the limited power to detect associations between the SNPs with the lowest MAF and IA in our cohort. The table below illustrates the required sample size (for an identical number of cases and controls) to obtain a power of 80% with a type I error below 5%.

MAF	Sample size		
	Relative risk, 1.5	Relative risk, 2.0	Relative risk, 2.5
0.01	7,963	2,398	1,247
0.05	1,689	911	272
0.10	911	283	151
0.15	657	208	113
0.20	535	172	95
0.25	466	152	85

Although there is indeed a low power for detecting associations, our primary rationale for selecting the SNPs was published evidence of their consequences to NOD2 function in host immunity as the result of their nonsynonymous variation, and associations with other inflammatory and infectious diseases.

- Did the investigators solely genotype polymorphisms in IL-10 and NOD2 in the clinical cohort of HSCT patients? If so, what was the rationale for selecting these genes and not other ones? If not, issues regarding correction for multiple testing should be addressed. This is particularly relevant as there is no replication in another cohort to confirm the genetic association.

The IL-10 (published elsewhere) and NOD2 SNPs were genotyped following a candidate gene approach and not a genome-wide scan. These genes were selected based on evidence for an

important role of NOD2 (current manuscript) and IL-10 (Carvalho et al., in preparation) in animal models of experimental aspergillosis and therefore, a biological plausibility for their involvement in susceptibility to infection in patients at-risk. Furthermore, the reason to select the NOD2 gene was based on previous evidence that NOD2 might played a role in the host response against *A. fumigatus* (as detailed in the introduction lines 91 -118)

Because our study did not involve a genome-wide scan where p-values of 0.01-0.05 would indeed be likely to pop up by chance, multiplicity correction is not typically required (especially in a study considering only 4 SNPs in NOD2). More important, and despite there is no replication for the association test results involving NOD2 in another cohort, we do present extensive functional validation by describing the functional consequences of the associated SNP with IA to the immune response to *Aspergillus* as well as data in animal models with genetic deficiency in NOD2.

We would also like to point out our newly included data that demonstrates that the carriers of the associated NOD2 polymorphism have lower levels of cytokines in the BAL during *Aspergillus* infection. (see new figure 1C)

- What was the method used for genotyping ? Was it part of a large-scale (or genome-wide scan) or just a few candidate genes ? Could the authors provide the MAF for the selected SNPs in the population studied and values for HWE. Is rs2066842 in linkage disequilibrium with rs20666847 (a LD table for R2 and D' might be useful).

We used a candidate gene approach and not a genome-wide scan. Genotyping was performed using KASPar assays (LGC Genomics, Hertfordshire, UK) in an Applied Biosystems 7500 Fast Real-Time PCR system (Thermo Fisher Scientific, MA, USA). Mean call rate was >97% for all genotyped SNPs. Quality control for the genotyping results was achieved with negative controls, common and rare homozygous controls (whenever available), and retesting of samples with indeterminate results. This information has been added to the Methods section of the manuscript (lines 636-647).

The table below illustrates the MAF in our cohort and values of HWE for the selected SNPs. (we updated the SNP table1 in our manuscript with the two additional columns.

RefSNP	Genome coordinates	aa change	Alleles	CEU MAF	MAF in our study	HWE
rs2066842	chr16:50710713	P268S	C>T	0.102	0.278	0.72
rs2066844	chr16:50712015	R702W	C>T	0.014	0.027	0.77
rs2066845	chr16:50722629	G908R	G>C	0.005	0.002	1.00
rs2066847	chr16:50729867	1007fs	->C	0.006	0.022	0.98

Following the reviewer's suggestion, we are also enclosing a table depicting LD (included as Table S1). rs2066842 can be defined to be in LD with rs2066847 based on the D' values. We also included this table in the supplementary information. The low r2 value between SNPs implies however a low predictability of the allele in the second locus and vice versa, and this is most likely explained by the low frequencies.

Table – LD analysis of the NOD2 SNPs evaluated in our study.

RefSNP	rs2066842	rs2066844	rs2066845	rs2066847
rs2066842	D'=1 r ² =1	D'=1 r ² =0.163	D'=1 r ² =0.044	D'=1 r ² =0.067

rs2066844	D'=1 r ² =0.163	D'=1 r ² =1	D'=1 r ² =0.002	D'=1 r ² =0.002
rs2066845	D'=1 r ² =0.044	D'=1 r ² =0.002	D'=1 r ² =1	D'=1 r ² =0.001
rs2066847	D'=1 r ² =0.067	D'=1 r ² =0.002	D'=1 r ² =0.001	D'=1 r ² =1

- Most patients undergoing HSCT receive mold-active prophylaxis, which may be administered for different duration depending of several factors. Is the association still significant when accounting for such prophylaxis in a time-dependant way ?

Although we currently do not have exact information on the duration of antifungal prophylaxis in our cohort, the established protocols indicate that antifungal prophylaxis should be typically administered from the start of conditioning until neutrophil recovery after transplant. Because only four cases of IA developed during the neutropenic period (and hence under prophylaxis), we reason that the association is not affected by prophylaxis. It is also noteworthy that prophylaxis was not associated nor tended toward risk of IA in univariate analyses, and was therefore not accounted for in the multivariate analyses.

- Was the association still significant when patients with infections due to mold other than *Aspergillus* accounted for ?

In our cohort, only five patients had infection with molds other than *Aspergillus* spp. (specifically, *Mucor* spp. and *Rhizomucor* spp.). By including these patients in the analyses, the association remains significant (the cumulative incidence for donor P268S was 33.4% for CC vs. 22.6% for CT and TT genotypes combined; p=0.03) (see figure below). While in theory, we could argue for a role of NOD2 SNPs in susceptibility to general mold infection, because the number of patients infected with molds other than *Aspergillus* is small, we believe that, at this point, it is more accurate to describe the association with IA only.

Genetic variation in donor P268S (rs2066842) and risk of mold infection after HSCT. Shown are the results obtained in a cohort comprising 315 eligible transplant donors. Cumulative incidence of mold infection according to a dominant genetic model of donor genotypes at rs2066842. Data were censored at 24 months, and relapse and death were considered competing events. P values were calculated using Gray's test.

Were the Crohn's disease patients providing samples under immunosuppressive treatment, as this might have artificially influenced the analyses.

The reviewer raises an important point, indeed immunosuppressive therapy may have influenced our analysis and may explain why phagocytosis and killing is not significantly enhanced. We discussed this in our discussion section lines 488-490. We were able to obtain the details on therapy from 5 of the nine patients and included this information in our methods section lines 630-634

Reviewer #3 (Aspergillus, mouse infection models)(Remarks to the Author):

An interesting paper in which the role of the NOD2 in host defence against pulmonary aspergillosis is investigated. Interestingly although NOD2 detects peptidoglycan, not present in *Aspergillus*, some data implicates it in *Aspergillus* immunity. NOD2 may however recognize chitin.

This is an excellent suggestion by the reviewer, therefore we explored in pilot experiments whether chitin, like the NOD2 ligand MDP, can reduce fungal killing by human or murine macrophages. Because this would provide a direction to further explore clear mechanistic insights of the enhanced fungi clearance in NOD2 KO mice. We were, however, unable to find an effect of chitin on the fungal killing capacity. See graph below

We agree that there are indeed studies suggesting that NACHT-LRR receptors (NLRs) such as NOD2 recognize chitin/chitosan. We mention this in our discussion (lines 513-519) Chitosan, a deacetylated form of chitin was found to activate the NLR, NOD-like receptor family, pyrin domain containing 3 (NLRP3) and thereby activate the inflammasome and induce IL-1 β production, whereas chitin did not activate NLRP3 (*Bueter 2014 J. Immunol*). Chitin itself was also found to induce IL-10 dependent on TLR9, mannose receptor and NOD2 (*Wagner 2014 PLoS Pathogens*). However in recent studies conducted by our group we demonstrated that chitin-induced IL-1Ra production in human PBMCs is independent of NOD2 (*Becker 2016 mBio*). In addition, this study demonstrates that chitin can synergize with the NOD2 ligand MDP to augment IL-1 β and TNF α responses, similar to how we observe that *Aspergillus* synergizes with MDP stimulation. We included this additional information in our discussion (lines 520-526)

The authors first show that SNPs in NOD2 influence probability to IA in HSCT. This was in the donor suggesting a defect in epithelial sensing rather than myeloid sensing.

Indeed the SNPs influence the probability to IA only if they are within the donor. However, this argues against a defect in epithelial sensing rather than myeloid sensing since the donor bonemarrow after transplantation will become the recipients immune system. This is why we discuss on lines 426-436 that most likely the myeloid compartment is responsible for the reduced susceptibility and that this is in line with our observations that NOD2 in myeloid cells, including murine BMDMs, human MDMs and human monocytes influences fungal killing capacity and phagocytosis.

SNP P268S CC increased risk of IA. Whilst I am not an expert in GWAS studies the data appear to be clear from the figures provided. However I do note the group have shown a lot of associations with a small cohort when compared to other GWAS studies. This could partly be because in immunosuppression there are a number of subclinical polymorphisms that could become relevant thus increasing the number of relevant SNPs, however I would advise a bioinformatition assess this data. For instance is there a correction for multiple testing?

This was not a genome-wide study, but a candidate gene approach targeting 4 SNPs in the *NOD2* gene. Therefore, multiplicity correction is not required. See also reply to reviewer 2.

The authors then go on to analyse PBMC responses to *A. fumigatus* in individuals with these SNPs. The authors find that the TT genotype of P268S results in less cytokine production. Therefore it appears that low risk for IA in HASCT for the TT genotype correlated with low cytokine production. **Indeed we observed reduced cytokine responses in individuals carrying the P286S TT genotype. We have now additionally analyzed cytokine levels in the BAL of aspergillosis patients. In line with the data in healthy volunteers we observed lower cytokine levels in patients carrying the P286S TT genotype. These data are included in figure 1C.**

The impact on fungal killing was only evident in a frameshift, resulting in increased fungal killing. **We were only able to identify an impact on fungal killing with the frameshift polymorphism, but it should be noted that due to the availability of a limited cell number these killing assays were performed in PBMCs. PBMCs, however, may not be the ideal cell population to assess fungal killing capacity due to the fact that PBMCs are a mixture of various immune cells of which numerous do not directly contribute to fungal killing. This may be an explanation to why we were not able to find an augmented killing for the P268S TT Genotype. We are aware that this is a limitation of our assessment of fungal killing using SNPs, but we were limited due to the availability of low numbers of cells.**

We have performed additional experiments to strengthen the conclusion that *NOD2* deficiency results in increased killing and have pinpointed that phagocytosis is increased in the absence of *NOD2*. We hope the reviewer can appreciate that these additional and previous data highlights a negative role of *NOD2* on the fungal killing capacity of human monocytes and macrophages, as well as murine BMDMs (Figure 7A-C and Figure 8A).

The authors then assessed the impact of *NOD2* deficient frameshift alleles on *A. fumigatus* responses. These cells had reduced cytokines and reduced CD4 T cells. To correlate findings, *NOD2* deficient mice were immunosuppressed with cyclophosphamide then infected with *A. fumigatus*. Under these conditions there was improved survival of *NOD2* $-/-$ mice. Bioluminescence demonstrated increased burden in WT mice. The *NOD2* $-/-$ mice had less evidence of invasive aspergillosis on histopathology although I would have liked to have seen some objective quantitative analysis of histopathology inflammation with imaging software if possible.

We have performed objective quantitative morphometric analysis using ImageJ. For morphometric analysis, fields at a magnification of x50, covering the entire lung sections of WT and *Nod2*^{-/-} mice at day2 post infection were analysed using ImageJ software. For each animal, we used the software to count the number of lesion foci per lung section, considering ischemic necrosis foci for wild-type mice and small macrophage infiltrates for *NOD2*^{-/-} mice. Using ImageJ software, we also calculated the size of ischemic necrosis and macrophage infiltrate foci. We have clarified this in our methods section (lines 792-796). The results are reported in Figure 5B.

The authors then hypothesize that the reduced susceptibility of *NOD2*^{-/-} may be due to increased killing capacity. *Nod2*^{-/-} BMDM had increased killing, however the authors do not show why.

From previous studies (Gresnigt 2018 J Innate Immunity) we are aware that the fungal killing capacity of macrophages can rely directly on the phagocytosis efficiency. Therefore, we have performed additional experiments in which we assessed phagocytosis of FITC-labeled *A. fumigatus* conidia. We were able to demonstrate that *Nod2*^{-/-} murine BMDMs as well as human MDMs in which *NOD2* was silenced show an enhanced capacity to engulf *A. fumigatus* conidia. Conversely, we observed that *NOD2* stimulation with the agonist MDP reduces the engulfment of FITC-labeled *A. fumigatus* in human MDMs and monocytes, but not in monocytes of *NOD2* deficient patients. These data is included as figure 7A-C and figure 8A.

The authors then propose that the protective effect of *NOD2* dysfunction is probably due to reduced inflammation, however this does not entirely make sense as other inflammatory PRRs confer susceptibility to aspergillosis. I think it is disappointing a mechanistic explanation is not found. The paper is otherwise novel and interesting, and will influence the field.

We thank the reviewer for noticing the novelty that will influence future directions in the field.

Indeed most other PRRs confer resistance to aspergillosis. The fact that NOD2 deficiency positively influences phagocytosis and fungal killing hinted us towards a more efficient recognition of the fungus. Therefore, we investigated expression of important PRRs post NOD2 stimulation, and observed a reduction of Dectin-1 expression upon NOD2 activation. Similarly, we observed an enhanced dectin-1 (*CLEC7A/Clec7A*) gene expression in human macrophages in which NOD2 was silenced as well as *Nod2*^{-/-} murine BMDMs.

Why NOD2 deficiency results in a reduced inflammatory response is difficult to explain. Either NOD2 itself recognizes fungal PAMPs and induces cytokine responses or NOD2 modulates expression of other PRRs (as discussed on lines 512-532).

We observed that NOD2 stimulation synergized with *Aspergillus* stimulation to boost *Aspergillus*-induced cytokine responses, similarly to how the fungal component chitin was previously demonstrated to synergize with MDP-mediated NOD2 activation to produce higher cytokine responses (see Becker et al 2016 mBio and discussion lines 519 - 525). Therefore one can argue that a yet unknown trigger for NOD2 present when *Aspergillus* is encountered by immune cells will augment cytokine responses, and in the setting of NOD2 deficiency this augmentation is not present and therefore results in lower overall cytokine production. Interestingly this is in clear contrast with NOD1 deficiency which results in increased cytokine production in response to *Aspergillus* making NOD2 a unique regulator in *Aspergillus* host defense that opens up a field of research to explore further in fungal infection as highlighted by the reviewer.

I was not clear what background the NOD2^{-/-} mice were in, not completely clear in methods. Otherwise possible to replicate the experiments.

The *Nod2*^{-/-} mice were from a C57Bl/6 background. We included this information in our methods section (line 738)

Figure S1 has no title.

In the revised manuscript we provide a title for figure S1

Reviewers' comments:

Reviewer #1 (Remarks to the Author):

The authors have performed several additional experiments and have addressed several of my (and other reviewers') comments.

The enhanced phagocytosis and killing (associated with an enhancement of dectin-1) of myeloid cells in the NOD2-deficient setting appears to be a plausible mechanism of the phenotype observed; data are provided in mice and humans and the mouse data of survival/fungal load corroborate such a potential mechanism. The cytokine data are less convincing as a potential mechanism, there are at times increased and at times decreased in different settings (humans/mice; different cell types) and the cytokine data in humans are not corroborating the SNP (Figure 1A/B) that the association with IA was found.

Reviewer #3 (Remarks to the Author):

The authors provide a detailed point by point review of the manuscript and add significant extra data. The key issue for this manuscript now is whether there is a clear mechanism that can be described for why NOD2^{-/-} leads to enhanced killing. It seems to be phagocytosis dependent but not clearly due to effects on Dectin-1.

Figure 1c please state P values somewhere

In figure 4 the authors measure cytokines in BAL. In accordance with the human studies they employ a neutropenic mouse model. Whilst this is entirely reasonable, it may be difficult to isolate innate defects in a model with such an established profound innate immune defect due to drugs. In addition, the measurements are at 3 days, which is rather late in animal model terms. Inflammatory airway responses are present from 6 hours post infection and often key differences are only seen early before second order variables kick in.

The initial human phagocytosis data is questionable for publication as there are only 2 data points. Please describe P values in Figure 7.

With regard to Figure 7, the literature relating to the importance of dectin 1 for phagocytosis of *Aspergillus fumigatus* is conflicting and probably only clear in dendritic cells. Accordingly the authors further investigations provide conflicting data, reinforcing the likelihood that this is not a dectin-1 dependent mechanism.

Point to point replies to Reviewers' comments:

Reviewer #1 (Remarks to the Author):

The authors have performed several additional experiments and have addressed several of my (and other reviewers') comments.

The enhanced phagocytosis and killing (associated with an enhancement of dectin-1) of myeloid cells in the NOD2-deficient setting appears to be a plausible mechanism of the phenotype observed; data are provided in mice and humans and the mouse data of survival/fungal load corroborate such a potential mechanism. The cytokine data are less convincing as a potential mechanism, there are at times increased and at times decreased in different settings (humans/mice; different cell types) and the cytokine data in humans are not corroborating the SNP (Figure 1A/B) that the association with IA was found.

We thank the reviewer for the appreciation of the additional mechanistic insights provided by our phagocytosis experiments and enhanced dectin-1 expression.

We agree that differences in cytokine production are a less convincing mechanism for the observed change in susceptibility upon NOD2 deficiency or genetic variation. We therefore have rewritten our discussion and interpretation of the results on this matter (Please see lines 764-782)

Reviewer #3 (Remarks to the Author):

The authors provide a detailed point by point review of the manuscript and add significant extra data. The key issue for this manuscript now is whether there is a clear mechanism that can be described for why NOD2^{-/-} leads to enhanced killing. It seems to be phagocytosis dependent but not clearly due to effects on Dectin-1.

Figure 1c please state P values somewhere

We have included all p-values in the figure.

In figure 4 the authors measure cytokines in BAL. In accordance with the human studies they employ a neutropenic mouse model. Whilst this is entirely reasonable, it may be difficult to isolate innate defects in a model with such an established profound innate immune defect due to drugs. In addition, the measurements are at 3 days, which is rather late in animal model terms. Inflammatory airway responses are present from 6 hours post infection and often key differences are only seen early before second order variables kick in.

The reviewer raises an important point that 3 days post infection may not be the ideal timepoint for assessment of cytokine responses, due to secondary effects setting. This is also why we raised this point in the discussion of the previous version of the manuscript.

We believe that our *in vitro* experiments, however, provide a more reliable view on the role of NOD2 in cytokine responses against *Aspergillus*.

Due to the fact that the murine *in vivo* cytokine data can be anticipated as confusing, and the fact that we indeed used a suboptimal timepoint of 3 days post infection, rather than an early timepoint such as 6 hours post infection, we decided to exclude this data from the manuscript to avoid confusion to the reader.

The initial human phagocytosis data is questionable for publication as there are only 2 data points. Please describe P values in Figure 7.

We agree with the reviewer that the initial phagocytosis data has a very preliminary nature with only two patients and 4 controls. We have included data in the first revision of three different *in vitro* models based on comments by the reviewers that provide robust evidence that NOD2 negatively influences phagocytosis. These new data make the initial data redundant. Therefore we decided to exclude the data of only 2 data points from the manuscript.

P-values in figure 7 are shown as asterixes (p<0.05 = *, p<0.01 = **, p<0.001 = *)**

With regard to Figure 7, the literature relating to the importance of dectin 1 for phagocytosis of

Aspergillus fumigatus is conflicting and probably only clear in dendritic cells. Accordingly the authors further investigations provide conflicting data, reinforcing the likelihood that this is not a dectin-1 dependent mechanism.

Indeed the role of dectin-1 and phagocytosis has been primarily assessed within the context of DCs and DC maturation. Notably some other studies have investigated dectin-1 in the context of phagocytosis within murine macrophages (Luther et al 2007 Cell Microbiol) and human monocytes (LC3-associated phagocytosis; Kyrmizi et al 2013 J. Immunol). The fact that we observe increased dectin-1 expression simultaneously with increased phagocytosis in the setting of NOD2 deficiency, and decreased phagocytosis simultaneously with decreased dectin-1 expression within the setting of NOD2 stimulation, may argue for a role for dectin-1 in this process.

Nevertheless, we agree with the reviewer that our data are not conclusive and we are aware that there may be other mechanisms at hand. Therefore we have rewritten our discussion section (lines 935-938).

REVIEWERS' COMMENTS:

Reviewer #1 (Remarks to the Author):

The authors have addressed my comments adequately